# Ectomycorrhizal access to organic nitrogen mediates $CO_2$ fertilization response in a dominant temperate tree

Peter T. Pellitier [1,2 ✉], Inés Ibáñez [1], Donald R. Zak [1 ✉], William A. Argiroff [1] & Kirk Acharya[1]

Plant–mycorrhizal interactions mediate plant nitrogen (N) limitation and can inform model projections of the duration and strength of the effect of increasing $CO_2$ on plant growth. We present dendrochronological evidence of a positive, but context-dependent fertilization response of *Quercus rubra L.* to increasing ambient $CO_2$ ($iCO_2$) along a natural soil nutrient gradient in a mature temperate forest. We investigated this heterogeneous response by linking metagenomic measurements of ectomycorrhizal (ECM) fungal N-foraging traits and dendrochronological models of plant uptake of inorganic N and N bound in soil organic matter (N-SOM). N-SOM putatively enhanced tree growth under conditions of low inorganic N availability, soil conditions where ECM fungal communities possessed greater genomic potential to decay SOM and obtain N-SOM. These trees were fertilized by 38 years of $iCO_2$. In contrast, trees occupying inorganic N rich soils hosted ECM fungal communities with reduced SOM decay capacity and exhibited neutral growth responses to $iCO_2$. This study elucidates how the distribution of N-foraging traits among ECM fungal communities govern tree access to N-SOM and subsequent growth responses to $iCO_2$.

[1] School for Environment and Sustainability, University of Michigan, Ann Arbor, MI, USA. [2] Department of Biology, Stanford University, Stanford, CA, USA. ✉email: ptpell@stanford.edu; drzak@umich.edu

Gross primary productivity (GPP) has been globally stimulated by rising anthropogenic $[CO_2]$[1,2] and Earth system models (ESM) suggest this effect could continue to *ca*. 2070[3]. While global-scale studies infer a moderate historical fertilization effect[1,4], evidence for rising $CO_2$ stimulating productivity at the ecosystem scale in mature forests has proven elusive[5–7]. This incongruity has limited accurate constraints of the fertilization effect in ESM[3], which is critical to predicting terrestrial carbon feedbacks that may continue to mitigate anthropogenic emissions[8,9]. Manipulative $CO_2$ enrichment experiments in mature forests record positive, albeit modest, and saturating growth responses to elevated $CO_2$ ($eCO_2$)[5,7,10–12]. Tree-ring studies investigating long-term plant biomass response to increasing ambient $CO_2$ ($iCO_2$) generally observe a weak fertilization effect[6,12–16]. These observations stand in contrast to the significant responses observed in early successional ecosystems[10,12,17], suggesting that any stimulatory effect of $CO_2$ may be transient. Limited nitrogen (N) availability, particularly in mature forests is widely implicated to constrain experimental and ambient growth responses to $CO_2$[10,12,18].

Plant N limitation is typically linked to the availability of inorganic N, which is made available via the microbial mineralization of soil organic matter (SOM)[19]. In contrast, N organically bound in soil organic matter (N-SOM), by far the largest ecosystem pool of soil N[20], is generally considered inaccessible to plants and is very rarely modeled to contribute to plant N budgets in ESM[21]. However, there is renewed interest in the possibility that acquisition of N-SOM may allow certain plants to "short-circuit" limiting supply rates of inorganic N[22,23]. In fact, estimates predict that acquisition of N-SOM, in addition to inorganic N sources, is necessary to mediate a sustained and positive plant growth response to $eCO_2$[24–27]. Owing to the fact that existing supply rates of inorganic N may be insufficient for a sustained fertilization response[28], projections of GPP under $iCO_2$ may be improved by considering the joint contribution of N-SOM and inorganic N to plant growth.

Plant acquisition of N-SOM is contingent on the activity of ectomycorrhizal (ECM) fungal symbionts[23,29]. ECM fungi may acquire N-SOM using enzymatic and non-enzymatic decay mechanisms retained from their free-living saprotrophic ancestors[23,30]. Despite their well-established role in providing plants with the majority of their annual N[29], ECM communities and their N-foraging traits have rarely been studied in relation to plant response to $iCO_2$[24,26,31,32]. Instead, ECM fungi are often implicitly treated as functionally equivalent in their capacity to decay SOM, leading to untested predictions that all ECM-associated host plants will be fertilized by $iCO_2$[24].

Biological market perspectives emphasizing the metabolic cost of fungal resource capture[33] suggest that plants may associate with ECM mutualists that maximize N acquisition and minimize plant carbon (C) expenditure (N return on C investment)[34,35]. ECM taxa vary widely in their capacity to decay SOM[30,36], with greater decay capacity likely carrying a greater C cost to their plant host[37]. Accordingly, ECM acquisition of N-SOM may be favored under conditions in which inorganic N availability is low[38]. For example, *Cortinarius* is a widespread ECM genus with substantial decay capacity[39] and is often associated with inorganic N poor soils[40]. In contrast, ECM taxa that consistently occur in high inorganic N availability soils may specialize in inorganic N acquisition[41,42]. Although the distribution of ECM taxa are also subject to complex community assembly processes[43], we reason that trees associating with ECM communities with greater decay potential (i.e. occurring in inorganic N poor soils) exhibit the largest relativized fertilization response to $iCO_2$ because both N-SOM and inorganic N contribute to tree growth (Fig. 1). In contrast, trees that primarily obtain inorganic N (i.e. occurring in inorganic N rich soils), exhibit a lower relativized growth response to $iCO_2$ (Fig. 1).

Tree rings represent the outcome of plant growth and are proxies for plant productivity[44]. Although not without methodological limitations[45], dendrochronological studies can be used to study historical responses to $iCO_2$[6,14–16,46]. This study had three major goals, (i) elucidate the potential contribution of N-SOM to tree growth, (ii) quantify ECM fungal community aggregated decay traits (CADT)[47] along a broad natural soil inorganic N gradient, and (iii) determine if CADT and tree N-SOM acquisition are linked with historical growth responses to $iCO_2$. To accomplish these coupled goals, we studied even-aged (~100 year old) *Quercus rubra* L. (northern red oak) individuals and their associated ECM communities occurring along a landscape-scale forest mosaic (~50 km) that forms a continuous soil inorganic N gradient (Supplementary Fig. 1)[48]. Bulk soil properties across this gradient are very similar, however, natural variation in soil inorganic N availability is derived from microsite differences in nutrient and water retention that have developed following glacial retreat ~10,000 years ago[48,49].

In this study we document that *Q. rubra* exhibits context-dependent growth responses to $iCO_2$. Overall, the largest relative responses occurred under conditions of low inorganic N availability, conditions where N-SOM is likely to contribute to tree growth; acquisition of N-SOM in these conditions is driven by the activity of specialized ECM fungal communities that are enriched in gene families associated with SOM decay. These results highlight the importance of plant-fungal interactions in mediating plant growth response to $iCO_2$ and suggests that ECM activity directly mediates plant access to large and poorly incorporated pools of N-SOM.

## Results

**Dendrochronological models and plant nitrogen uptake.** In order to evaluate the potential contribution of N-SOM to plant growth, we used a change point analysis[50] within a dendrochronological Bayesian modeling framework. We asserted that if tree growth, measured as basal area increment (BAI) ($cm^2/y$), is primarily constrained by inorganic N availability[51], we can predict a relatively consistent increase in annual growth along a soil gradient of net N mineralization (i.e., inorganic N supply) after accounting for other growth limiting factors (Fig. 2A). However, if tree growth (BAI) is supplemented with N-SOM, the relationship between observed growth rates and the supply of inorganic N should be weakened (smaller slope in Fig. 2A). BAI of focal trees from 1980 to 2017 was analyzed as a function of net N mineralization rates, tree size (to detrend size-age effects), growth in the prior year (to account for lag effects), and as a function of regional yearly climatic conditions (to reflect year to year environmental variability). Finally, BAI was modeled as a function of spatially explicit random effects to account for distances between all sampled focal trees and sites (see *Methods*). Average minimum May temperature displayed the highest correlation with BAI, whereas annual variability in seasonal and monthly precipitation were only weakly correlated with yearly estimates of BAI and were not included in the final model (*Methods*). Overall model fit was high ($R^2 = 0.94$; Fig. S2), and we detected a change point in modeled plant growth (BAI) along the N mineralization gradient at $0.53 \pm 0.01$ μg inorganic N $g^{-1}$ $d^{-1}$ (mean ± SD; Fig. 2B).

Consistent with our predictions, there was as significant change-point in the relationship between BAI and rates of net N mineralization, the slope of which was significantly weaker (smaller) to the left of the change point than it was to the right (Fig. 2C; 95% PIs did not overlap). Notably, tree growth below the BAI statistical change-point was significantly greater than predicted based on the

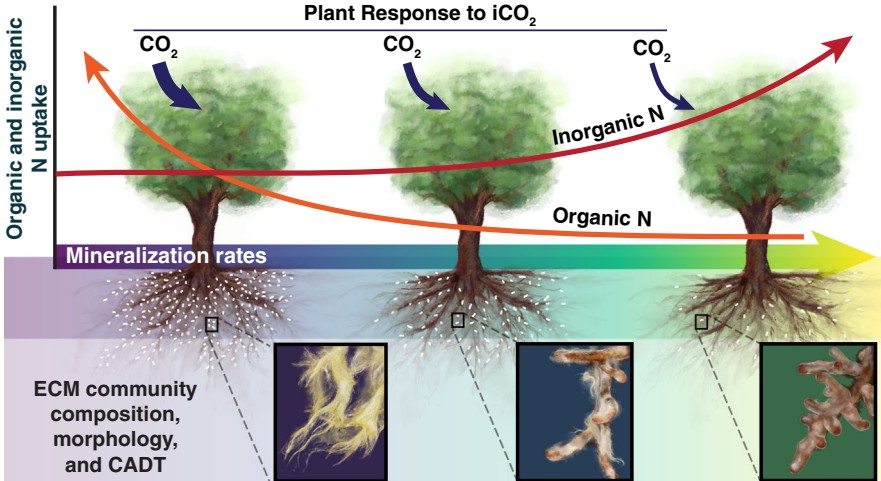

**Fig. 1 Hypothesized contribution of different nitrogen (N) forms to tree growth (red and orange lines; y-axis) and tree responses to historic increases in $CO_2$ (i$CO_2$).** These responses occur along a gradient of varying supply rates of inorganic N (net N mineralization rates: x-axis). Dark blue arrows show the relativized fertilization response (arrow width) to i$CO_2$. ECM fungal community composition, morpho-traits, and community aggregated decay traits (CADT) estimated using metagenomic approaches, are hypothesized to vary with soil inorganic N availability. Note hypothesized turnover in the dominance of ECM taxa with extra-radical rhizomorphic hyphae and long-and medium-distance exploration morphologies. White speckles on roots depict hypothesized relative abundance of ECM root-tips. Illustration by Callie R. Chappell, with permission from Reinhard Agerer.

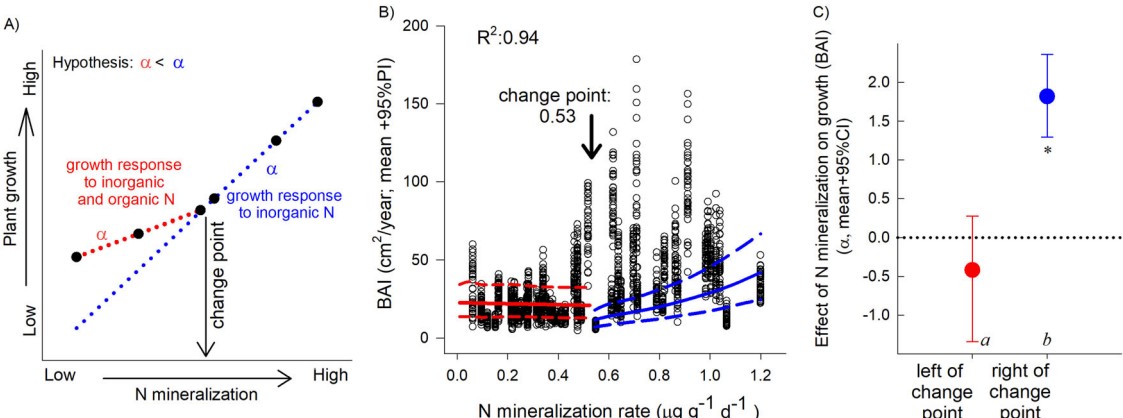

**Fig. 2 Analysis framework and modeled contribution of inorganic N and N-SOM to plant growth. A** Representative analysis of tree growth as a function of net N mineralization rates. The change point analysis identifies the occurrence and location of an inflection point, if any, and the value of the slope parameters on each side. **B** Basal Area Increment (BAI), from 54 *Q. rubra* trees along the studied net N mineralization gradient (black circles correspond to an individual growth year). Red and blue lines indicate model estimated BAI mean and 95% PI above and below the identified change-point (BAI change point; estimates were calculated at average values of the other covariates). $R^2$ denotes overall Bayesian model fit. **C** Slope parameters are significantly different from each other (95% CIs do not overlap; different letters (*n* = 54). Asterisks indicates parameter is different from zero (95% CI does not overlap with zero).

contribution of inorganic N alone. After accounting for other growth-limiting variables and in accordance with data presented below, we interpret this higher than predicted growth response as the putative contribution of N-SOM to tree growth.

**Ectomycorrhizal community analyses.** Concomitant with modeled BAI responses to soil inorganic N availability, there was significant turnover in the composition and morphological attributes of ECM communities. These shifts were generally consistent with the hypothesized context-dependent contribution of N-SOM and inorganic N to tree growth (Fig. 3; Supplementary Fig. 3). The ECM genus *Cortinarius* dominated in conditions of low inorganic N availability (Fig. 3A)[52]. *Cortinarius*, has been widely hypothesized to participate in the acquisition of N-SOM using a potent repertoire of oxidative enzymes that rivals certain free-living saprotrophic fungi[42,53]. In contrast, the ECM genus

*Russula* which is associated with inorganic N or labile organic N acquisition[54], occurred in high relative abundance in soils with high inorganic N availability (Fig. 3B). Other ECM taxa, such as the white-rot derived *Hebeloma* was found at relatively higher abundance in low inorganic N soils and may also contribute to the functional shifts observed here[52,55] (Supplementary Fig. 4). We also examined the relative abundance of ECM taxa forming rhizomorphic hyphae and extraradical emanating hyphae, morpho-traits associated with acquisition of organic N[37,56,57]. The relative sequence abundance of ECM genera forming rhizomorphic hyphae ($F_{1,110} = 11.32$, $P = 0.001$) and medium-distance exploration types ($F_{1,110} = 14.65$, $P = 0.0002$), was significantly greater for *Q. rubra* individuals occurring below the statistical BAI change point than in soils above it (Fig. 3C, D).

In addition to taxonomic and morphological assessments, we employed shotgun metagenomic sequencing of ECM communities

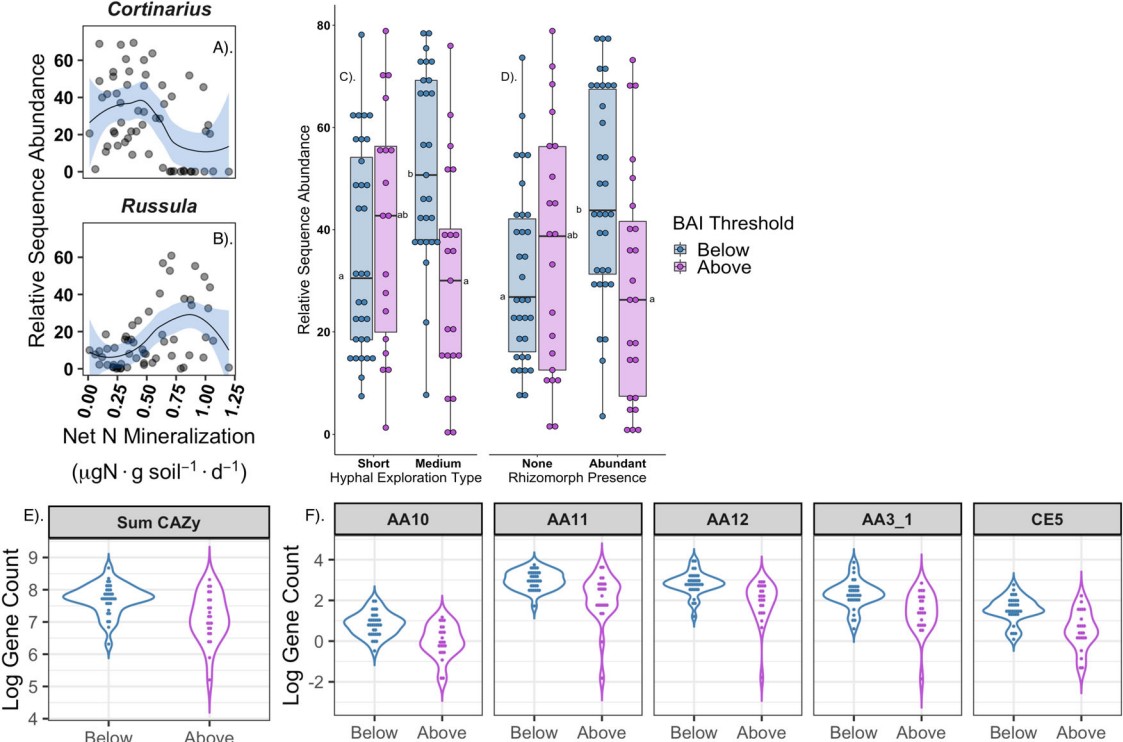

**Fig. 3 Compositional, morphological and functional turnover along the soil nitrogen gradient consistent with shifts in ectomycorrhizal (ECM) fungal N foraging traits. A**, and **B** relative sequence abundance of the ECM fungal genera *Cortinarius* and *Russula* along the gradient of net N mineralization rates (x-axis). Colored bands depict GAM fits. **C** Box-and whisker plot depicting ECM fungi forming short or medium-distance exploration types. Letters denote statistical significance **D**. ECM fungi forming rhizomorphic hyphae above and below the BAI statistical threshold (0.53 µgN g soil$^{-1}$ day$^{-1}$); letters adjacent to median line of box, denote statistical significance, points are individual communities totaling $n = 58$. Upper and lower hinges depict 25$^{th}$ and 75$^{th}$ percentiles. **E** Sum of CAZy gene counts ($n = 100$ gene families). **F** Specific gene families (headers) significantly enriched below the BAI change point. AA10, AA11 encode lytic polysaccharide monooxygenases, AA12 is an oxidoreductase, AA3_1: cellobiose dehydrogenase, CE5: acetyl xylan esterase.

inhabiting each of the focal *Q. rubra* individuals to estimate ECM community aggregated decay traits (CADT)[47] and the potential contribution of N-SOM to tree growth. We removed non-fungal sequences and annotated sequences encoding enzymes mediating the decay of SOM[47], using the Carbohydrate Active Enzyme and Redoxibase databases (100 gene families studied; Table S2)[58,59]. We accounted for the compositional nature of the metagenomic data using near-single-copy gene standardization (*Methods*) and scaled CADT by the number of colonized ECM root-tips encountered on each tree (Supplementary Figs. 5 and 6), reasoning that the net quantity of decay genes hosted on *Q. rubra* roots is necessary to link N-SOM with host uptake (see *Methods*).

*Q. rubra* individuals that exhibited growth responses consistent with N-SOM uptake, i.e., most individuals occurring in low inorganic N soils, hosted ECM communities with distinct decay profiles (Supplementary Figs. 7 and 8). Moreover, these ECM communities were significantly enriched in the total abundance of CAZy genes ($F_{1,54} = 8.38$, $P = 0.0006$; Fig. 3E). Several key gene families that were significantly enriched below the statistically derived BAI change point include cellobiose dehydrogenase (AA3_1) and lytic polysaccharide monooxygenases (LPMO's: AA9, AA10, AA11) which together act to decay SOM[60,61] (Fig. 3F; Supplementary Fig. 9). Many other gene families that are speculated to participate in the decay of SOM also exhibited significant shifts (Supplementary Fig. 9; Supplementary Table 3). Generalized dissimilarity model (GDM) analyses revealed that the compositional abundance of ECM decay genes exhibited a sharp community level threshold response to rates of net N mineralization at approximately 0.5 µg inorganic N g soil$^{-1}$ day$^{-1}$ (Supplementary Fig. 9). This threshold response is congruent

with the independently derived BAI change point (Fig. 2B), and suggests that ECM communities below this threshold have similar genomic decay potential across a range of soils with low inorganic N availability. Below the modeled BAI change point, differential transcription of certain ECM gene families may explain the greater potential contribution of N-SOM to tree growth with progressively decreasing inorganic N availability. In support of this potential explanation, ECM decay genes are known to be under tight transcriptional regulation, with the expression of key oxidative enzymes inversely correlated with soil inorganic N availability[42].

**Tree growth response to increasing CO$_2$.** The same 38 years of ring data were used to estimate the relative response of plant growth (BAI) to iCO$_2$. During the sampled growth period (1980–2017), atmospheric CO$_2$ increased by ~70 µmol mol$^{-1}$ (www.NOAA.gov), serving as a natural gradient of iCO$_2$. To compare growth rates along the N mineralization gradient, we calculated annual estimates of growth N efficiencies (GNE) for each tree, $GNE = \frac{BAI}{N \, mineralization}$. These estimates were standardized, and then analyzed as a function of yearly atmospheric [CO$_2$], average minimum May temperature, and spatially explicit random effects (Fig. 4A, D; see *Methods*). Slope parameters (λ) (i.e., iCO$_2$ effect) for individual trees were then analyzed as a function of net N mineralization rates using a change point method (Fig. 4B, E). Model estimates derived a change point in the relationship between tree growth and the effect of CO$_2$ (λ) along the N mineralization gradient at a value of 0.39 ± 0.01 µg inorganic N g$^{-1}$ d$^{-1}$ (Fig. 4E; Supplementary Fig. 2). Trees

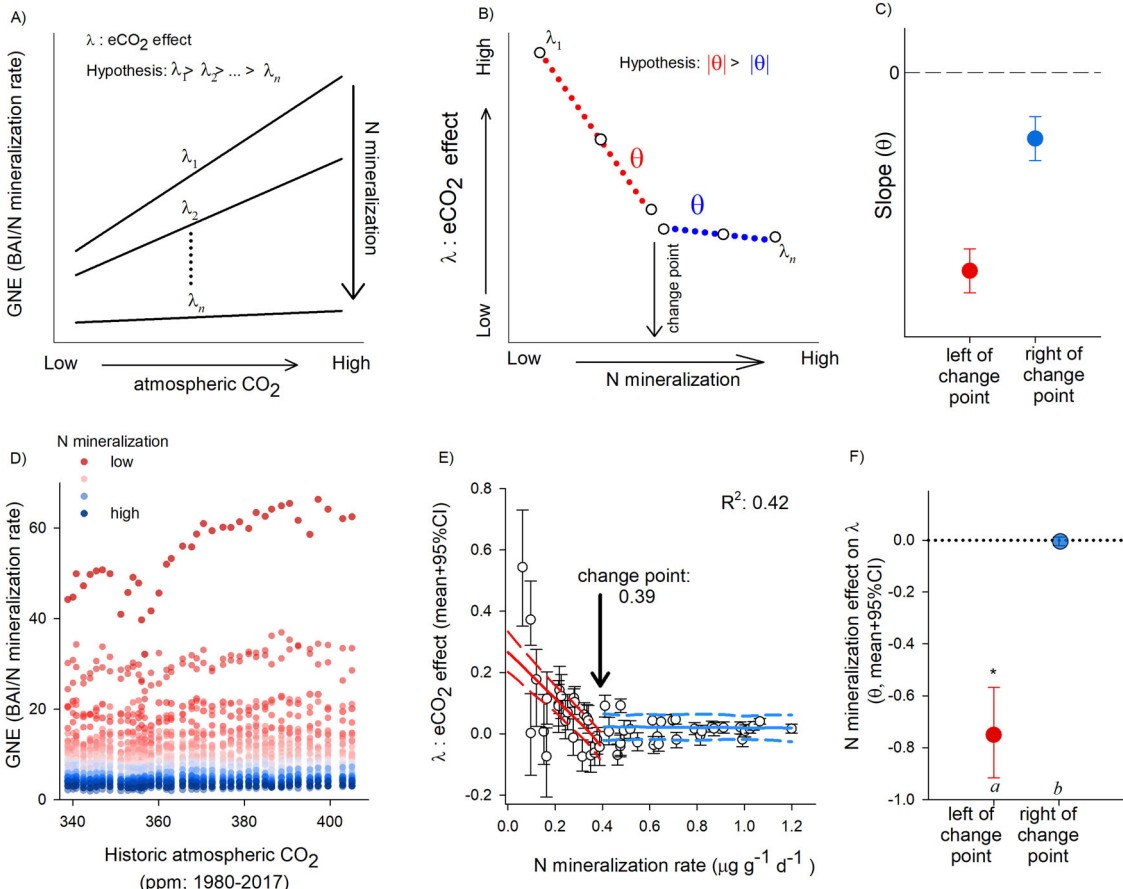

**Fig. 4 Analysis framework and evidence for context-dependent iCO₂ fertilization responses. A** Representative analysis framework of growth-nitrogen efficiency index (GNE) as a function of increasing concentrations of historic atmospheric $CO_2$ at each point along the net N mineralization gradient (different lines and their relative slopes). **B** Conceptual diagram of the effects of iCO₂ (λ) on plant growth; change point analysis can detect an inflection point along the soil gradient, if any. **C** Differences in θ derived from the red and blue portion of panel B, indicate distinct slope values. **D** Dendrochronological data collected from 54 *Q. rubra* trees from the past 38 years. Individual points represent estimated annual GNE values colored by tree-specific rates of net-N mineralization. **E** Individual points represent individual trees response to iCO₂ (mean model slopes derived from **D**) over the study period. Red and blue lines denote Bayesian change-point model with plotted 95% PI (dashed lines; estimated at average values of other covariates) (n = 54). **F** Denotes mean and 95% CI for the red and blue slopes depicted in **E**; different letters denote significant differences between slopes, asterisks indicate significant differences from zero (95% CI do not include zero; n = 54).

growing in soils below this statistical change point tended to have increasing GNE with iCO₂ concentrations, resulting in a positive CO₂ fertilization response. In contrast, trees inhabiting soils with greater inorganic N availability (to the right of the statistical change point) displayed no changes in GNE over the past 38 years (slopes reflecting the effect iCO₂ on plant growth were not different from zero; Fig. 4F). The modeled slope parameters (θ), to the left and right of the change point were significantly different from one another (Fig. 4F).

## Discussion
Acquisition of N-SOM has been widely implicated as a primary determinant of plant response to rising $CO_2$[12,24,26,27], but has been challenging to disambiguate in forest ecosystems.

We present evidence suggesting that plant acquisition of N-SOM is contingent on the community N-foraging traits of ECM fungi and secondly that N-SOM contributes to a positive plant growth response to iCO₂. In support of our hypotheses, tree growth responses to iCO₂ were context dependent and primarily positive for *Q. rubra* individuals where N-SOM was likely to contribute to tree growth. In contrast, trees occupying soils with relatively high inorganic N availability, modeled as reliant on inorganic N alone, displayed no positive growth response to iCO₂.

The near convergence of the change-points derived from both dendrochronological analyses (BAI & GNE), supports the role of N-SOM in stimulating a positive CO₂ fertilization response. Furthermore, the synchronous shifts in ECM community composition, morpho-traits associated with organic N acquisition and metagenomic estimates of SOM decay capacity (CADT) largely aligns with the independent dendrochronological models, supporting a strong role of ECM fungi in modulating *Q. rubra* N acquisition and growth responses to iCO₂. Additional evidence for a transition from assimilation of N-SOM to predominately inorganic N economies comes from a previous study of these individuals trees, which found δ15N depleted foliage under conditions of low inorganic N availability suggestive of enhanced organic N uptake, and relatively enriched foliage under high inorganic N availability[52]. Overall, assumptions of plant reliance on inorganic N alone, or alternatively, ubiquitous plant access to N-SOM via ECM symbionts, are each unable to resolve the context dependent growth responses to iCO₂ observed here.

Our investigation of ECM communities builds towards a novel perspective on the contribution of N-SOM to plant growth by investigating possible N-foraging trait trade-offs along a soil nutrient gradient[38]. Shifts in the morphological and genomic traits of ECM communities across the soil mineralization gradient

could result from niche based processes and metabolic tradeoffs in the cost of foraging for N-SOM *vs.* inorganic N (N return on C investment)[38]. For example, ECM communities inhabiting trees in low inorganic N soils were enriched in C intensive hyphal morphologies (rhizomorphs and medium-distance exploration types) associated with organic N acquisition[37]. These traits may be especially favored under conditions of low inorganic N availability if they are associated with enhanced photosynthate allocation and persistence on host roots[38]. Moreover, these ECM communities were enriched in key gene families that together form a lytic polysaccharide monooxygenase (LPMO)[61] decay pathway. Notably, previous laboratory studies have shown that ECM fungi express LPMO to decay SOM[62]. While certain CAZy gene families studied here are very likely implicated in SOM decay, many CAZy have additional intracellular roles or are involved in mycorrhizal initiation, and can therefore be challenging to demonstratively ascribe to SOM decay[61,63]. Manganese peroxidase is a potent oxidative enzyme that certain ECM fungi employ to degrade SOM[42]; our results demonstrate that the presence of this gene family decreases across the breadth of the gradient, however, it did not exhibit a significant threshold response to the BAI change-point. Comparative analyses that link the enzymatic acquisition of N-SOM with transcriptomic community-level ECM profiles and plant uptake, would substantially bolster the mechanisms proposed here, but remain technically infeasible under field conditions[30]. Additionally, metagenomic estimates of absolute functional gene abundance may provide deeper insight into the N-cycling pathways and mechanisms proposed. Future studies studying co-occurring saprotrophic communities may reveal interesting inter-guild interactions that structure plant access to N and SOM dynamics[64].

Our results support predictions asserting that supply rates of inorganic N are insufficient to engender a sustained positive plant growth response to rising $CO_2$[28,65]. This may partially explain the results of certain tree-ring studies finding minimal effect of iCO$_2$ on the accumulation of plant biomass[6,14,16]. Moreover, this interpretation is consistent with evidence from Free Air $CO_2$ Enrichment (FACE) experiments that document negligible effects of eCO$_2$ on plant growth unless additional sources of N are added, typically in the form of inorganic N[17,66,67]. In the current study, tree association with ECM communities with enhanced capacity to obtain N-SOM may have increased plant N assimilation beyond that made possible by endogenous supply rates of inorganic N alone. In contrast for trees with high existing growth rates, greater C allocation towards ECM symbionts that specialize on inorganic N acquisition may not translate to greater plant N uptake and positive responses to iCO$_2$. This is due to the fact that inorganic N release is mediated by free living soil saprotrophs[19], whose activity is largely independent from ECM fungi. In contrast, progressive increases in C allocation to ECM communities with high N-SOM acquisition potential, could provide a direct mechanism for plants to enhance N uptake under iCO$_2$ conditions[25,68], but see[69]. Accordingly, the responses we describe are consistent with previous observations that identify greater N uptake as necessary for positive responses to $CO_2$, and not merely from greater N use efficiency[70]. We note that the addition of inorganic N in certain previous studies is likely to disfavor ECM decay of SOM[54], in these cases precluding accurate assessment of the role of N-SOM in stimulating tree growth under eCO$_2$. Finally, although a recent study suggested that ECM acquisition of N-SOM may reduce soil C stocks under rising $CO_2$ conditions[27], our results are not consistent with evidence of asymmetric soil C depletions.

While the evidence we have accumulated provides strong support for the role of ECM fungi and N-SOM in mediating plant response to iCO$_2$[25], it simultaneously challenges expectations that ECM associated trees will display a positive growth response to rising $CO_2$[24,26,32]. Because the positive growth responses to iCO$_2$ reported here are relativized and primarily occurred for the individuals with the smallest initial BAI, our study suggests that positive biomass responses to iCO$_2$ may be modest, and if they can be extrapolated, suggests that ECM associated tree response to iCO$_2$ is overestimated[28,32]. In fact our findings overall range from a modest to neutral growth response to iCO$_2$ which are in line with available evidence for eCO$_2$ experiments conducted in mature ecosystems[7]. Together, our work further highlights the importance of nutrient limitation in iCO$_2$ responses, suggesting that studies derived from aggrading or early successional ecosystems may be poor predictors of forest C sequestration potential under future $CO_2$ regimes[12]. Further studies on a wider range of ECM and AM communities and host trees, distributed across forest biomes, particularly the tropics, are needed to determine the generality of our findings.

Our field-based study is not without certain caveats. To address some of them, our modeling framework directly accounted for historic increases in temperature, thereby allowing us to isolate the effect of iCO$_2$ on individual-level plant growth. It is also possible that other limiting nutrients such as phosphorus (P)[32] could account for the distinct growth patterns found here. This is unlikely, however, as prior study of soils in this region show small differences in P availability[71]; moreover, P is unlikely to be a key limiting nutrient in these young soils (<10,000 yrs). Similarly, previous studies in this region also suggest that free-living and symbiotic N-fixation are unlikely to provide additional sources of inorganic N that we did not measure[72]. While historic rates of N deposition are inaccessible at the scale of individual focal trees or plots studied here, they are also unlikely to vary significantly across plots, as our study area is small and uniformly distant from anthropogenic point sources of N pollution. Additionally, the relative differences in inorganic N availability among forest plots studied here have remained stable since at least the mid 1980's[48,49], and the minimal disturbances in this late-successional ecosystem also suggests relative equilibrium in the distinct N-cycling pathways. A key consideration of our proposed mechanism is that N-SOM has differentially contributed to plant growth for the duration of the study period. While we cannot directly test this assumption, previous analyses of ECM communities occupying these soils suggest that communities have remained relatively stable, supporting this possibility[73]. Further study on the mechanistic role of rhizosphere priming under iCO$_2$ conditions is needed[31], but alone does not seem capable of driving the context-dependent results reported here. Finally, we acknowledge that there are considerable biases associated with sample design that affect the interpretation of dendrochronological data such as measuring only a subset of dominant trees in an area which if non-randomly selected can easily bias towards the preferential selection of large diameter age classes[45]. We address some of these issues by employing relativized analyses of host trees, selected at random, at small geographic scales[45].

Plant productivity responses to iCO$_2$ remain one of the largest uncertainties in projections of the terrestrial C sink[8,9,74]. We provide dendrochronological evidence of a context-dependent stimulation of tree growth by iCO$_2$ in a mature temperate forest ecosystem; this response is putatively driven by the uneven contribution of N-SOM to tree growth. Functional trait trade-offs in the N-foraging attributes of ECM fungal communities may govern plant growth responses to iCO$_2$; this finding provides important nuance to predictions asserting that all ECM associated trees will respond positively to iCO$_2$. Moreover, we highlight that singular emphasis on inorganic N in existing ESM

may lead to spurious conclusions regarding the strength and duration of the $CO_2$ fertilization effect[28,75]. In these respects, our findings add further evidence that projections of substantial increases in terrestrial C storage driven by $CO_2$ fertilization of GPP are likely overestimated[76]. While incorporation of ECM communities into existing ESM is not feasible, further work will determine if inorganic N availability can represent a reliable proxy for plant access to N-SOM in ecosystems dominated by ECM fungi. In conclusion, the accurate representation of plant growth response to $iCO_2$ and the terrestrial C sink in ESM appears to be contingent on incorporating N-SOM, a previously unrecognized and cryptic source of plant N.

## Methods

**Site conditions**. Our study took place across a regional network of 12 forest sites comprising a natural N soil gradient in Manistee National Forest in northwestern Lower Michigan (Supplementary Fig. 1). These sites have been described in full elsewhere[52]. Briefly, annual rates of net N mineralization range from 38 to 120 kg N ha$^{-1}$ y$^{-1}$ (calculated from[48]), which broadly spans soil inorganic N availability in the upper Lake States region[77,78]. The forest stands are even-aged (~100 years old), resulting from regrowth following forest clearing in the early 20th century. Relative differences in soil nutrient availability have persisted for decades due to lack of disturbance[48,49]. Micro-site climatic differences in nutrient retention have developed in the past ~10,000 yrs resulting in variation in nutrient cycling (Supplementary Fig. 1). An interlobate moraine transects the study region, and a network of outwash plains with slightly coarser textured soils are associated with lower rates of net N mineralization; in contrast, soils occurring in more upland positions on moraines are associated with greater rates of net N mineralization. Soils across the study region are derived from sandy (~85% sand) glacial drift, and range from Typic Udipsamments to Entic Haplorthods. Comparison with previously published rates of net N mineralization for soils from this region suggest that regional variation in inorganic N availability has persisted since at least the mid 1980's[48,49], and likely across their historical development. Moreover, comparison of rates of net N mineralization revealed that the underlying inorganic N gradient studied here is stable across the duration of the growing season (Pearson $r = 0.77$; Supplementary Fig. 10).

**Tree core sampling and measurement**. In May 2018, at each of the 12 sites, we randomly selected five mature *Q. rubra* individuals that were at least 10-m apart, and measured tree diameter at breast height (1.3 m; DBH) in cm. We then extracted growth cores (to the pith) from the North and South aspect of each tree at DBH using 5.15 mm Haglöf increment borers. The samples were dried overnight at 100 °C. We mounted tree cores on cradles and progressively sanded them by hand, from 100 to 600 grit. The mounted cores were digitized on a flatbed scanner at a resolution of 1200 dpi. We measured yearly ring width (growth) of the scanned tree cores using the Cybis CooRecorder program at a precision of 0.001 mm (Cybis Elektronik 2010). We then used the program Cybis CDendro for individual cross dating and chronology assembly by site. Crossdating was achieved when TTest values were greater than 5 for matching target samples using the P2YrsL normalization method[79]. We created master ring width lists that were summed by stem to reflect the average yearly growth of each individual using cross dated North & South aspects. We estimated historical DBH of focal trees at each year using the yearly ring width from our master chronologies. All overstory plant stems greater than 10 cm DBH and within 10 m radius of each focal *Q. rubra* individual were measured and identified (Supplementary Figs.: 11-12). Understory plant communities are reported elsewhere[52], and no N-fixing taxa in the overstory or understory were encountered.

**Characterization of soil properties**. Five soil cores, 5-cm diameter and 10-cm deep, were collected both May and August 2018, were taken radially around the dripline of each focal *Q. rubra* individual following our previous work in these forests[52]. Soil net N mineralization rates were quantified as an estimate of soil inorganic N availability for soil samples collected in both May and August 2018. Soil inorganic N was extracted from fresh sieved soil using 2 M KCl, followed by a 14-day aerobic incubation in order to measure rates of soil inorganic N mineralization[80]. $NO_3^-$ and $NH_4^+$ in soil extracts were analyzed colorometrically (AQ2; Seal Analytical, Mequon, WI) (Supplementary Methods). Total free primary amines (TFPA) in soil (primarily amino acids and amino sugars) was measured using fresh sieved soil extracted with 2 M KCl[81] (Supplementary Fig. 12). Total C and N and soil pH were processed as described in[52].

**Analysis of BAI response to N mineralization rates**. Annual tree radial growth and diameter at breast height (DBH) at the time of sampling were combined to calculate past DBH by subtracting radial growth each year to the previous year's DBH. We then calculated annual tree basal area (BA), BA = π(DBH/2)$^2$ and basal area increments (BAI; cm$^2$/y). BAIs were estimated for each tree ($i$) and year ($y$) as

the difference in BA between two consecutive years:

$$BAI_{i,y} = BA_{i,y} - BA_{i,y-1}. \quad (1)$$

Fifty four trees yielded high quality ring widths for BAI estimates. We then analyzed these BAI estimates as a function of tree-level N mineralization rates applying a change point analysis that would identify if at any point in the N mineralization gradient the relationship with growth changed (Fig. 2A; Supplementary Table 5). We also accounted for the relationship between tree size (DBH) and growth by including DBH in the analysis of BAI (detrending). Furthermore, because tree growth at any particular year may also be affected by lag effects (growth in previous years)[82] we included previous year growth (standardized) as a covariate after exploring how many years back previous growth could have affected current growth, and a lag of one showed the best relationship (Table S5). We carried out exploratory data analysis to identify climatic variables (monthly temperature and precipitation records from the closest NOAA climate station in Cadillac, MI [data was retrieved June 7, 2020, https://www.ncdc.noaa.gov/cdo-web]). Average May minimum temperature was the variable with the highest correlation with BAI (r: 0.11); we included this variable in the analysis (standardized). To also account any spatial autocorrelation, e.g., trees close to each other and site level factors, we included spatially explicit random effects (SERE). For any particular tree $i$ and year $y$ BAI analysis likelihood and process models were:

$$BAI_{i,y} \sim \log Normal(B_{i,y}, \sigma_{i,y}^2) \quad (2)$$

$$B_{i,y} = (\alpha_1 + J_i\alpha_2) + (\alpha_3 + J_i\alpha_4) \cdot Nminer_i + \alpha_5 \cdot \ln(DBH_{i,y}) + \alpha_6 \cdot BAIS_{i,y-1} + \alpha_7 \cdot MayTemp_y + SERE_i \quad (3)$$

$$\sigma_{i,y}^2 = a + b \cdot \ln(DBH_{i,y}) \quad (4)$$

We followed a Bayesian approach to estimate parameters. Parameter $J$ is an indicator, with value 0 before the change point, and value 1 after. This change point parameter was estimated as: *Change point* $\sim Uniform(0, 1.25)$, allowing the change point to fall outside the range of N mineralization sampled (0.06–1.19 µg g$^{-1}$ d$^{-1}$). Variability around growth estimates ($\sigma_2$) was estimated as a function of DBH, since this seems to vary with size[83]. The rest of the parameters were estimated from non-informative prior distributions. In Eq. (6), d represents the distance between sites $i$ and $j$.

$$a \sim \log Normal(1, 1000) \quad (4)$$

$$\alpha_*, b \sim Normal(0, 10000) \quad (5)$$

$$SERE_i \sim Exponential\left(\sum_{i=1}^{N} e^{-\varphi d_{ij}}, \sigma_{SERE}^2\right), \quad (6)$$

$$\varphi \sim Uniform(0.001, 10) \quad (7)$$

$$1/\sigma_{SERE}^2 \sim Gamma(0.0001, 0.0001). \quad (8)$$

**Characterization of Ectomycorrhizal fungi**. In August 2018, ECM root-tips were collected radially around the dripline of each focal *Q. rubra* individual as previously described[52]; briefly, five cores were taken around each tree, each core was 10-cm deep and 11 × 11 cm in area. The soil was removed from roots using sequential washing using tap water and ECM root-tips with high turgor were manually excised using a dissecting microscope after visually eliminating non-*Quercus* roots. In total, 14,944 individual ECM root-tips were excised. DNA was extracted from lyophilized root-tips using the Qiagen DNeasy Plant Mini Kit (Hilden, Germany) and DNA pools were split for amplicon and metagenomic sequencing (see below). The ITS2 fragment of rRNA was amplified using PCR, following Taylor et al.[84] (Table S1) and sequenced using Illumina Mi-Seq (2 × 250; San Diego, CA). Sequences were processed using DADA2 v1.16, ASV were assigned taxonomy using the UNITE dynamic database (v.8; 97–99% sequence similarity) with the scikit naive bayes algorithem (v.0.21.0)[85,86]. We used the DEEMY (characterization and DEtermination of EctoMYcorrhizae) database (http://www.deemy.de/) to gather morphological information on the exploration type (hyphal foraging distance) and rhizomorph occurrence of ECM taxa present in our dataset at >0.5% relative abundance[52]. We assigned morphological hyphal trait data for 28 ECM genera, comprising more than 93% of all identified ECM sequences.

**Metagenomic sequence generation, processing, and annotation**. Shotgun metagenomic sequencing was conducted using a NovaSeq 6000 instrument (2 × 150 bp) at the University of Michigan Advanced Genomics Core. In total, 23,203,326,006 metagenomic sequences were generated, and reads were left unmerged. In order to remove non-fungal sequences, we removed sequences that mapped to the UniVec database (bacterial, archaeal, human, viral) sequences, as well as *Quercus rubra*[87] and *Qurcus lobata* genome assemblies[88] using Kraken2 (v. 0.9.29)[89]. On average, 22% of sequences per sample were removed during this filtering step, yielding a mean of 307,041,274 putative fungal sequences per sample (Fig. S12). Next, we used a direct mapping approach to annotate remaining sequences against the CAZy and Peroxibase reference databases (100 total gene families; Table S2) using 'sensitive' DIAMOND (v. 0.9.29)[90] and BWA-MEM

(v.0.7.17)[91] respectively, following best practice for unmerged reads[92]. The compiled decay gene database primarily contained 'core' gene families found to be actively expressed during fungal decay of SOM and microbial biomass[93,94] (CAZy: http://www.cazy.org; http://peroxibase.toulouse.inra.fr/), and is hereafter referred to as the 'CAZy database' (Table S2). We tabulated the number of near-single copy genes, as a proxy for the number of Dikaryotic fungal genomes present in each sample, using the OrthoDB v.9 gene database, which comprised 1,312 near-single copy gene variants[95]. Further methodological details are presented in the *Supplementary Methods*.

**Statistical analysis of Ectomycorrhizal fungal composition and metagenomic function**. We compared the relative sequence abundance of ECM ASV forming rhizomorphic and medium-distance hyphal morphologies above and below the statistical BAI change point using two-way ANOVA. To account for the compositional nature of the metagenomic decay gene data[96] we calculated the logarithm of the number of sequences mapped to a given decay gene family divided by the geometric mean number of orthologous near single-copy features present in the sample (single-copy genes); note that this is identical to an additive log-ratio transformation[96]. We incorporated potential underlying shifts in the biomass of ECM communities by multiplying single-copy standardized gene counts by the standardized number of colonized ECM root-tips recovered from focal *Q. rubra* root-systems (Supplementary Fig. 5). To isolate the effect of mineralization rate and other environmental variables in driving shifts in the compositional abundance of the 100 decay gene families, we used generalized dissimilarity models (GDM)[97,98]. Environmental predictors initially included in the model were net N mineralization rates, pH, soil C and N, C:N, TFPA and gravimetric soil moisture. All abiotic measurements were calculated at the individual tree basis. This model additionally incorporated geographic distances between individual focal trees and Bray-Curtis distance matrices of the abundance of the 100 gene families were used. We used backwards model selection[99], and confirmed the significance of remaining predictors using matrix permutation (*nperm* = 500) (Table S4). To determine the identity of the ECM gene families that were significantly enriched in communities inhabiting trees that exhibited putative uptake of N-SOM, we compared the mean log abundance of individual gene families occurring in ECM communities above and below the statistical BAI changepoint using one-way ANOVA with Bonferroni correction (Table S3). *Vegan* (v 2.5.6)[100] and *tidyverse* (1.3)[101] were also employed for analysis.

**Analysis of BAI response to atmospheric $CO_2$**. To compare trends in growth across trees we first calculated an index of growth nitrogen efficiency, GNE. This index was estimated for each tree (i) and year (y) as:

$$GNE_{i,y} = \frac{BAI_{i,y}}{Nminr_{\cdot i}}. \tag{9}$$

To facilitate the analyses, we standardized GNE:

$$GNES_{i,y} = (GNE_{i,y} - \overline{GNE_i})/SD_{GNEi} \tag{10}$$

Afterwards, we analyzed GNES as a function of annual atmospheric $CO_2$ (obtained from NOAA 2019), average May minimum temperature (see above), and spatially explicit random effects (Table S6-7). For each tree *i* and year *y*:

$$GNES_{i,y} \sim Normal(G_{i,y}, \sigma_i^2) \tag{11}$$

$$G_{i,y} = \beta_i + \lambda_i \cdot CO_{2y} + \gamma_i \cdot MayTemp_y + SERE_i \tag{12}$$

We then analyzed the effect of $CO_2$, slope parameters λ (mean and variance, $\bar{\lambda}_i$ and $\sigma_\lambda^2$), as a function of N mineralization rate; this analysis tested how much of the variability found in this parameter could be attributed to differences in inorganic N availability. We tried several analyses, including exponential decay and logarithmic functions, and change point analysis with two different intercepts. The best model fitting the data, based on Deviance Inference Criterion (DIC)[102] was a simple change point analysis (Table S6-7). For each tree *i*:

$$\bar{\lambda}_i \sim Normal(L_i, \sigma_\lambda^2) \tag{13}$$

$$L_i = (\theta_1 + J_i \cdot \theta_2) + (\theta_3 + J_i \cdot \theta_4) \cdot miner_i \tag{14}$$

Parameter *J* is an indicator, with value 0 before the change point, and value 1 after. This change point parameter was estimated as: *Change point* ∼ *Uniform*(0, 1.25). Remaining parameters were estimated from non-informative prior distributions.

$$\beta_*, \gamma_*, \theta_* \sim Normal(0, 10000) \tag{14}$$

$$SERE_i \sim Exponential\left(\sum_{i=1}^{N} e^{-\varphi d_{i,j}}, \sigma_{SERE}^2\right) \tag{15}$$

$$1/\sigma_{SERE}^2 \sim Gamma(0.0001, 0.0001) \tag{16}$$

$$\varphi \sim Uniform(0.001, 10) \tag{17}$$

See supplement for additional information and all analysis code.

**Reporting summary**. Further information on research design is available in the Nature Research Reporting Summary linked to this article.

## Data availability
Raw DNA sequences associated with the ITS2 amplicon sequencing are deposited in NCBI Sequence Read Archive: SRR14164239-SRR14164298. Metagenomic sequences are deposited under accession codes: SRR15377920-SRR15377978. Associated soil metadata are available in Dryad (https://doi.org/10.5061/dryad.4f4qrfjbt). Access to wood cores will be available upon written request. Publicly available datasets used in this study include CAZy and Redoxibase http://www.cazy.org; http://peroxibase.toulouse.inra.fr/). Univec database: https://ftp.ncbi.nlm.nih.gov/pub/UniVec/. UNITE database: https://unite.ut.ee/. NOAA climatic data https://www.ncdc.noaa.gov/cdo-web]) was used, and in addition, OrthoDB database was used (https://www.orthodb.org/). Finally, the publicly available DEEMY database was accessed at http://www.deemy.de.

## Code availability
Custom code for dendrochronological analyses is reported in the Supplementary Methods.

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

## Acknowledgements

We especially thank R. Upchurch, E. Herrick, K. Seguin, N. Gudal, N. Ahmad and B. VanDusen valuable laboratory and field support. V. Denef, M. Coon, J. Evans and T. James provided essential sequencing and bioinformatic support.

## Author contributions

P.P., I.I. and D.Z. designed the study. P.P. and I.I. performed the analyses. W.A. and K.A. contributed intellectually to this project as well as to field and laboratory work. P.P drafted the paper, and all authors contributed to the final copy.

## Competing interests

The authors declare no competing interests.
