## [Peer Review File · Nature Communications]

Reviewer comments, first round -

Reviewer #1 (Remarks to the Author):

The manuscript Pellitier et al. presents an interesting study which tries to elucidate role of nitrogen limitation in plant productivity response to elevated CO₂. Without any doubts this is a very timely topic with significant impact on various research fields and potentially also our society. In general, I found the study well conducted and well written. However I have a few concerns which should be addressed before publication. I would also like to stress that I am a researcher in a field of fungal ecology and role of fungi in ecosystems, but I do not have any experience with dendrochronological research. Therefore I do not feel qualified to provide more profound evaluation of the aspects related to the methods and interpretation of the results of plant growth based on the dendrochronological data. Of course, I carefully read that part as well and I found it sound, but please keep in mind that my opinion is not an expert opinion.

Major concerns:

The authors employed shotgun metagenomics sequencing to link ectomycorrhizal fungal community and their enzymatic potential activity with tree uptake of N-soil organic matter (N-SOM). In general, the methodology was sound, but evaluation and presentation of the obtained data is not very clear to me. My major concern is related to Figure 4B. First of all, it is crucial to include data point or other kind of information of data variation. Presenting a function only looks suspicious. Besides that I am not sure that I fully understand the logic behind this graph. As far as I understood from the methods, the "partial ecological distance" was calculated as a Bray-Curtis distance of the abundance of the 94 selected gene families. If this is true then Figure 4B probably shows correlation between the Bray-Curtis pairwise distance and pairwise distance of the net N mineralization rate.

Many parts of results (e.g. lines: 199-203) provide discussion of the results and make the discussion redundant. Similarly, some parts of the results repeat method section (lines: 176-178).

Minor concerns:

Figure S1 should provide better understanding of the sampling site. I would recommend zooming in on the sampling sites. Including differentiation of the sites by colored based on the rates of net N mineralization would substantially improve the figure.

Lines 108-109: Is there a study which really tested and proved that EcM fungi with enhanced decay capacity carry a greater C cost to their host plant?

Figure 1: Please make size of the trees the same. Although it is described in the legend that the tree size is not informative, I find it unnecessarily confusing.

Line 426: Can you specify number of reads which mapped to Fungi?

Reviewer #2 (Remarks to the Author):

The authors use a novel combination of dendrochronology, biogeochemistry, and fungal ecology to show that oak trees growing in sites with low inorganic N availability harbor ECM fungi that have a greater potential to mine N from SOM and have growth rates that respond relatively more to elevated CO₂ than oak trees growing in sites with high inorganic N availability. Overall, the paper is well written and provides multiple lines of correlative evidence to support this main conclusion. Below, I highlight my main concerns followed by more detailed comments.

The evidence for ECM trees mining N from SOM in the low inorganic N sites is compelling but fails to meet the standards to document N mining from SOM that the corresponding authors highlighted in their 2018 paper in *New Phytologist*. The current manuscript only shows that the ECM fungi have the genetic potential for liberating N from SOM but does not show that these genes were transcribed, the enzymes liberated N, or the ECM fungi transferred N liberated to the host plant. While I do not think these standards are necessary for the inference the authors are making here, it would benefit the manuscript if the authors discussed why or why not these standards were applicable in the present study.

Pellitier, P.T. and Zak, D.R., 2018. Ectomycorrhizal fungi and the enzymatic liberation of nitrogen from soil organic matter: why evolutionary history matters. *New Phytologist*, 217(1), pp.68-73.

Further, I am wondering if there is additional evidence supporting this mining of N from SOM than simply the fact that trees below the N Mineralization threshold don't show growth responses to increases in inorganic N availability. Would it be possible to do a simple N budget (I recognize there would be a lot of uncertainty) to show that the N mineralization rates are not sufficient to support growth and that organic N access would be necessary to support NPP? Similarly, in the sites above the break point is there evidence that the trees are using all of the inorganic N mineralized? One other thing, it could also be interesting to speak to the fact that the trees relying most on N mining from SOM have the smallest N reserve to draw from and thus might not sustain a large response to elevated CO₂ in the future.

What was the rationale for treating the trees at the different sites as individual replicates instead of using the 12 sites as the replicate in the statistical analyses? It would also benefit the reader to understand why N mineralization varies so much across this area besides references to microclimate shifts and glacial formations. Can you be more specific on what is driving the gradient?

Also could you provide some additional site-level data including maybe average basal area of the trees, average BA per m², and more importantly the co-occurring species. Is red oak dominant at each of these sites or does the prevalence of AM trees increase as inorganic N increases? This would have important implications for interpreting the data and the overall message of the manuscript if the species composition of the sites shifts with N availability. As such, I think this data is necessary for the manuscript.

More detailed comments:

Figure 4 is clearly an outlier in the excellent figures presented in the manuscript. All of the other figures provide visual cues for interpreting and understanding the data presentation or are fairly intuitive. The heat map on the left and the curve in the right panel don't seem to add that much to the story and maybe could be put in SI.

Lines 86-89: This isn't entirely true. ESM mass balance C and N inputs and outputs in soils and this almost reads as if they don't get the magnitude of soil N correct. In these models, there is N that is in slow cycling SOM pools that plants can't access. It would be more accurate to state that large scale ESM's don't include a mechanism whereby plants can access N bound in SOM as well as inorganic N.

Lines 89-90 and the final line of the manuscript: The idea that plants short circuit the N cycle and access organic nitrogen forms is not cryptic or relatively unrecognized. This is at least a twenty-year old idea and may even be older (see Schimel and Bennett 2004 Ecology and references there-in). It might be better to frame this as ecosystem scale evidence or something else along those lines.

Line 101-102: This language makes it read as if this present study is an explicit test. It is compelling evidence but it is not an explicit test of this idea. See large comment above.

Line 231: This paper shows a CO₂ effect on tree rings:
Mathias, Justin M., and Richard B. Thomas. "Disentangling the effects of acidic air pollution, atmospheric CO₂, and climate change on recent growth of red spruce trees in the Central Appalachian Mountains." *Global change biology* 24, no. 9 (2018): 3938-3953.

Line 245: I understand the need to highlight the importance of the findings here but suggesting that growth responses to elevated CO₂ for ECM trees are likely modest based on 54 chronologies is a bit of an overreach.

Reviewer #3 (Remarks to the Author):

[Editor's note: this reviewer was asked specifically to evaluate the tree ring aspects, including the

core sampling and BAI calculations]

Dear authors,

I had been able to look at your submission "Ectomycorrhizal access to organic nitrogen mediates tree CO₂ fertilization response", which I consider a very thorough and comprehensive study. While I am not an expert in the field, I can say that the tree ring-related aspect of the study is sound and well done.

Reviewer #4 (Remarks to the Author):

I found the manuscript by Pellitier et al. confusing, overly complicated in conceptual framework, and poor in methodological treatment. Even though they use a Bayesian approach, tree ring studies are not suitable to robustly detect changes driven by CO₂ because it is almost impossible to isolate the CO₂ signal from the effects driven by hundreds of other environmental factors and their interactions.

- The discussion is poor. If I understood correctly, the main conclusion is that ECM plants use organic N when inorganic N is low. This seems pretty obvious. The other conclusion is that ECM plants do not increase their growth in response to CO₂ rising when inorganic N is high. This second conclusion contradicts probably the most well settled conclusion from hundreds of CO₂ experiments concluding that the CO₂ fertilization effect is high when N (inorganic N) is high (e.g. Reich et al. 2006, Norby et al. 2010, Norby & Zak 2011, Reich & Hobbie 2013). This seems to be the most important conclusion and yet it is not discussed by the authors. As a consequence, the only potentially surprising conclusion is simply ignored. This left me with the impression that the conclusions are not justified due to biases in the method (see below), so the authors try to hide it.

- Indeed, a number of studies have found no evidence in tree-ring widths for any positive effect of CO₂ on growth. However, the methods used to test for long-term trends in tree-ring records are flawed and "may introduce considerable biases in quantifications of forest responses to environmental change" (Nehrbass-Ahles et al. 2014 GCB). See also Brienen et al. 2012. This is vociferously acknowledged by the tree-ring community, and they recognize more work is needed to make tree-ring data useful as a method to study trends in plant growth with environmental drivers while accounting for confounding factors (e.g. tree age, size). As a consequence of this bias, findings of no growth stimulation with rising CO₂ (e.g. van der Sleen et al. 2014) are common, but inconsistent with all other evidence (Walker et al. 2020).

I believe these issues should not prevent tree-ring analyses from being published. The impact of such analyses for CO₂ effects is however doubtful, unless these biases are acknowledged and dealt with in an innovative way. To a bare minimum, these issues should be acknowledged in the intro, and the language should be humbler given the limitations. I don't support the bold language used in the intro, equating the validity of CO₂ experiments and tree-ring analyses in quantifying CO₂ effects.

- Their hypothesis doesn't make any sense to me: "we reason that if N-SOM is critical to a positive eCO₂ fertilization effect, then trees associating with ECM taxa with greater decay capacity, i.e. occurring in N poor soils, will exhibit the largest relative fertilization response to eCO₂ (Fig. 1). In contrast, we predict that trees that primarily obtain inorganic N, i.e. occurring in inorganic N rich soils, will exhibit a reduced growth response to eCO₂ (Fig. 1)". To my understanding, N limits the capacity of plants to convert eCO₂ to additional growth. It doesn't matter if the N comes from inorganic or organic sources. In any case, using organic sources would decrease growth, not increase. The reason is that, as the authors describe in the intro, plant organic N uptake requires a carbon investment in ECM fungal mining of N, which reduces the amount of carbohydrates that can be allocated to plant growth. I am puzzled with this hypothesis, which is the central argument of the paper. You can look at it from a different perspective: why would the CO₂ effect be larger under low than high N availability? It doesn't make any sense.

- Fig 1: similar to my comment right above, I don't understand the reasoning in Fig. 1. The figure is extremely confusing. The tree with the widest CO₂ arrow (representing larger CO₂ fertilization effect) is represented by the smallest tree size. Shouldn't be the opposite? Also, the CO₂ arrows

give the impression they refer to CO₂ concentration

- Intro: the statement linking the (lack of) response in mature eCO₂ experiments with results from tree-ring analyses is misleading; this statement suggests there is no such thing as a CO₂ fertilization effect, as corroborated by both mature experiments and tree-ring analysis. This is incorrect. The vast majority of sources of evidence suggest that CO₂ increases carbon uptake.

Tree-ring studies are the exception (e.g. Walker et al. 2020 New Phyt). It is true that the few CO₂ experiments with mature plants show no growth stimulation. However, there are just a handful or less of examples, and all of them occur in extreme nitrogen (Sigurdson et al.) or phosphorus (Ellsworth et al, Jiang et al) limitations and results from these experiments cannot be extrapolated to other ecosystems (comment by Luo & Niu 2020).

- The entire framework is too complicated, and the logic is hard to follow. For example, it took me a while to understand why the authors expect a relationship between BAI and N mineralization, which is the main foundation of their conclusions. Even after reading the paper, I am not too sure I follow the logic. I must say I have participated in over a dozen papers on the N cycle, and yet I struggled to follow. I doubt the topic, structure, methods and writing of the paper are suitable for a wide audience, as someone would expect in Nature Communications.

- L74 "models" x2

- I don't understand why GPP is mentioned in the abstract and is the lead in the introduction. The authors focus on stem growth, not GPP. NPP would be closer, but they just measure BAI which is far from NPP.

- L75 there is no consensus about the statement "[...] ca. 2010". That is just a result from a particular study and not a generalization. Stimulation of GPP by eCO₂ would continue as long as CO₂ continues increasing or the saturation point of photosynthesis is reached. Both points are uncertain, and the effect doesn't stop in a particular year. Please remove because it gives a wrong impression.

REVIEWER COMMENTS

Reviewer #1

The manuscript Pellitier et al. presents an interesting study which tries to elucidate role of nitrogen limitation in plant productivity response to elevated CO₂. Without any doubts this is a very timely topic with significant impact on various research fields and potentially also our society. In general, I found the study well conducted and well written. However I have a few concerns which should be addressed before publication. I would also like to stress that I am a researcher in a field of fungal ecology and role of fungi in ecosystems, but I do not have any experience with dendrochronological research. Therefore I do not feel qualified to provide more profound evaluation of the aspects related to the methods and interpretation of the results of plant growth based on the dendrochronological data. Of course, I carefully read that part as well and I found it sound, but please keep in mind that my opinion is not an expert opinion.

Response: Thank you for your careful and favorable read of our work.

Major concerns:

The authors employed shotgun metagenomics sequencing to link ectomycorrhizal fungal community and their enzymatic potential activity with tree uptake of N-soil organic matter (N-SOM). In general, the methodology was sound, but evaluation and presentation of the obtained data is not very clear to me. My major concern is related to Figure 4B. First of all, it is crucial to include data point or other kind of information of data variation. Presenting a function only looks suspicious. Besides that I am not sure that I fully understand the logic behind this graph. As far as I understood from the methods, the “partial ecological distance” was calculated as a Bray-Curtis distance of the abundance of the 94 selected gene families. If this is true then Figure 4B probably shows correlation between the Bray-Curtis pairwise distance and pairwise distance of the net N mineralization rate.

Responses: Figure 4 has been moved to the supplement. This ‘function’ represents model output from a generalized dissimilarity model (GDM). It is a standard means of presenting environmental and biotic drivers of compositional variation in amplicon and metagenomic dissimilarity across samples. It is similar to published works employing metagenomic analyses of environmental communities (Bouma-Gregson et al. 2019; ISME). We have now included two additional panels in Figure 2, presenting the ‘raw’ gene counts for each sample, for the sum of all examined gene families and those that displayed the greatest overall shifts above and below the dendrochronological change point.

Bouma-Gregson, K., Olm, M.R., Probst, A.J., Anantharaman, K., Power, M.E. and Banfield, J.F., 2019. Impacts of microbial assemblage and environmental conditions on the distribution of anatoxin-a producing cyanobacteria within a river network. *The ISME journal*, 13(6), pp.1618-1634.

Many parts of results (e.g. lines: 199-203) provide discussion of the results and make the discussion redundant. Similarly, some parts of the results repeat method section (lines: 176-178).

Response: We have thoroughly revised the entirety of the text to improve clarity and prevent repetition.

Minor concerns:

Figure S1 should provide better understanding of the sampling site. I would recommend zooming in on the sampling sites. Including differentiation of the sites by colored based on the rates of net N mineralization would substantially improve the figure.

Response: A new map (Figure S1) has been created to address this with satellite imagery and sampling sites colored with soil mineralization rates.

Lines 108-109: Is there a study which really tested and proved that EcM fungi with enhanced decay capacity carry a greater C cost to their host plant?

Response: Measuring carbon cost to the host plant represents a very challenging activity (*sensu* Bogar et al. 2019; Mycorrhiza). However the suggestion that these fungi carry a greater C cost to the host plant is consistent with available theory on nutrient foraging and expression of enzymes involved in SOM degradation (*sensu* Kirk and Farrell 1987: Applied Environmental Microbiology; Hobbie and Agerer 2010 Plant Soil). To address this concern, we have rewritten this statement to accommodate the relative lack of literature measuring this flux, line 57. Our statement however is consistent with previous interpretations and hypotheses:

“Because ECM taxa vary widely in their capacity to decay SOM^{31,34}, with those with greater decay capacity likely carrying a greater C cost to their plant host³⁵, ECM acquisition of N-SOM may be favored under conditions in which inorganic N availability is low”

Kirk, T.K. and Farrell, R.L., 1987. Enzymatic "combustion": the microbial degradation of lignin. *Annual Reviews in Microbiology*, 41(1), pp.465-501.

Bogar, L., Peay, K., Kornfeld, A., Huggins, J., Hortal, S., Anderson, I. and Kennedy, P., 2019. Plant-mediated partner discrimination in ectomycorrhizal mutualisms. *Mycorrhiza*, 29(2), pp.97-111.

Hobbie, E.A. and Agerer, R., 2010. Nitrogen isotopes in ectomycorrhizal sporocarps correspond to belowground exploration types. *Plant and Soil*, 327(1), pp.71-83.

Figure 1: Please make size of the trees the same. Although it is described in the legend that the tree size is not informative, I find it unnecessarily confusing.

Response: This has been fixed in an updated Figure 1.

Line 426: Can you specify number of reads which mapped to Fungi?

Response: Please see the sentence line 386. “On average, 22% of sequences per sample were removed during this filtering step, yielding a mean of 307,041,274 putative fungal sequences per sample”. We have also included a supplementary figure documenting that the number fungal sequences per sample does not vary along the studied soil gradient (Figure S17).

Reviewer #2

The authors use a novel combination of dendrochronology, biogeochemistry, and fungal ecology to show that oak trees growing in sites with low inorganic N availability harbor ECM fungi that have a greater potential to mine N from SOM and have growth rates that respond relatively more to elevated CO₂ than oak trees growing in sites with high inorganic N availability. Overall, the paper is well written and provides multiple lines of correlative evidence to support this main conclusion. Below, I highlight my main concerns followed by more detailed comments.

Response: Thank you for this feedback and favorable read.

The evidence for ECM trees mining N from SOM in the low inorganic N sites is compelling but fails to meet the standards to document N mining from SOM that the corresponding authors highlighted in their 2018 paper in *New Phytologist*. The current manuscript only shows that the ECM fungi have the genetic potential for liberating N from SOM but does not show that these genes were transcribed, the enzymes liberated N, or the ECM fungi transferred N liberated to the host plant. While I do not think these standards are necessary for the inference the authors are making here, it would benefit the manuscript if the authors discussed why or why not these standards were applicable in the present study.

Pellitier, P.T. and Zak, D.R., 2018. Ectomycorrhizal fungi and the enzymatic liberation of nitrogen from soil organic matter: why evolutionary history matters. *New Phytologist*, 217(1), pp.68-73.

Response: Thank you for your careful read of our previous work. We have now more thoroughly discussed the elements that are missing from our correlative study in the Discussion section. We agree that further study of ECM liberation and transfer of N-SOM represents an important mechanistic component for further study. However, these approaches remain very challenging under field conditions.

Specific wording on line 233:”Comparative analyses that link the enzymatic liberation of N-SOM with transcriptomic community-level ECM profiles and plant uptake, would substantially bolster the mechanisms proposed here, but remain technically infeasible under field conditions”³⁰”

Further, I am wondering if there is additional evidence supporting this mining of N from SOM than simply the fact that trees below the N Mineralization threshold don't show growth responses to increases in inorganic N availability. Would it be possible to do a simple N budget (I recognize there would be a lot of uncertainty) to show that the N mineralization rates are not

sufficient to support growth and that organic N access would be necessary to support NPP? Similarly, in the sites above the break point is there evidence that the trees are using all of the inorganic N mineralized?

Response: Evidence for plants obtaining N derived from SOM is in fact consistent with your recommendation. Plants occurring in low inorganic N soils show growth responses that are greater than predicted based on inorganic N availability. We agree that an N budget offers a useful framework to add additional insight into our findings. However, ecosystem pools for the necessary compartments are not available, precluding us from a complete itemization required for N budget analysis. Foremost, for an N budget to be useful in relation to our study, we would need paired N budgets for trees wherein ECM communities are relatively enriched and depleted in genes encoding enzymes degrading SOM. Such a comparison would need to occur at each site along the inorganic N availability gradient, enabling us to make direct comparisons. This situation does not occur in nature and we therefore cannot address this concern.

An additional line of evidence that supports plant uptake of N-SOM is derived from foliar isotopic findings presented in Pellitier et al. *Ecosystems in press* and attached in this review bundle. Red Oak foliage under conditions of low inorganic N availability has highly depleted foliage, consistent with ECM fractionation of N-SOM. In contrast, foliage above this breakpoint had enriched foliage, consistent with inorganic N uptake. We have now briefly discussed our previous work as an additional independent line of evidence supporting our findings.

Specifics on line 236: “Additional evidence for a transition from *Q. rubra* assimilation of N-SOM to predominately inorganic N comes from a previous study of these individuals trees, which found isotopically depleted foliage under conditions of low inorganic N availability and relatively enriched foliage under high inorganic N availability⁴⁸”

One other thing, it could also be interesting to speak to the fact that the trees relying most on N mining from SOM have the smallest N reserve to draw from and thus might not sustain a large response to elevated CO₂ in the future

Response: We have included a brief discussion of the implications of our results in Earth system models, in particular the longevity of this stimulatory effect. Throughout the Discussion section, you will find a more detailed discussion of our work in context with previous CO₂ experiments in forest ecosystems and tree-ring studies.

Specifically we have discussed this potential here on line: 193: “Because the positive growth response to iCO₂ reported here is relativized and primarily occurred for the individuals with the smallest initial BAI, our study suggests that positive biomass responses to iCO₂ may be modest, and if they can be extrapolated, suggest overestimates for ECM tree response to iCO₂ at global scales^{28,32}”

See also Line 270: Moreover, we highlight that singular emphasis on inorganic N in existing ESM may lead to spurious conclusions regarding the strength and duration of the CO₂

fertilization effect^{28,67}. In these respects, our findings add further evidence that projections of substantial increases in terrestrial C storage driven by CO₂ fertilization of GPP are likely overestimated⁶⁸.

What was the rationale for treating the trees at the different sites as individual replicates instead of using the 12 sites as the replicate in the statistical analyses?

Response: While the trees are considered individual replicates, in all of the dendrochronological models, geographic coordinates are used to calculate pairwise distances among trees, both within and among sites. In this way, geographic proximity among trees is considered as an explicit random effect. Moreover, the metagenomic generalized dissimilarity models (GDM) also explicitly consider geographic distances among trees, therefore taking into account the geographic correlation matrix. Furthermore, by using individual trees as our unit of analysis we were able to match each tree's growth with ECM community composition and function and soil inorganic N, and thus better account for the variability we found in the plant growth response variables. All measurements are conducted at the individual tree level, including all edaphic covariates.

It would also benefit the reader to understand why N mineralization varies so much across this area besides references to microclimate shifts and glacial formations. Can you be more specific on what is driving the gradient?

Response: Detail such as the following has been added to the main text in the methods section. Line 291. "An interlobate moraine transects the study region, and a network of outwash plains with slightly coarser textured soils are associated with lower rates of net N mineralization; in contrast, soils occurring in more upland positions on moraines are associated with greater rates of net N mineralization. Soils across the study region are derived from sandy (~85% sand) glacial drift, and range from Typic Udipsamments to Entic Haplorthods"

Also could you provide some additional site-level data including maybe average basal area of the trees, average BA per m², and more importantly the co-occurring species. Is red oak dominant at each of these sites or does the prevalence of AM trees increase as inorganic N increases? This would have important implications for interpreting the data and the overall message of the manuscript if the species composition of the sites shifts with N availability. As such, I think this data is necessary for the manuscript.

Response: This data has now been provided in the supplement (Figure S10-S11), and discussed in brief in the main text. Specifically, the proportion of each co-occurring overstory plant stem greater than 10cm, and within 10m of each focal *Quercus rubra* at greater than 1% relative abundance has been included as a supplement (Figure S10). Secondly, 'neighborhood' stem density has also been included and does not significantly vary across the studied soil gradient (Figure S11). Moreover, we note a previous study from this region found no N-fixing trees or understory plants are recorded within 10m of the focal *Q. rubra* individuals (Pellitier et al., *in press* Ecosystems).

Figure 4 is clearly an outlier in the excellent figures presented in the manuscript. All of the other figures provide visual cues for interpreting and understanding the data presentation or are fairly intuitive. The heat map on the left and the curve in the right panel don't seem to add that much to the story and maybe could be put in SI.

Response: We have moved this figure to the supplement based on your suggestion.

Lines 86-89: This isn't entirely true. ESM mass balance C and N inputs and outputs in soils and this almost reads as if they don't get the magnitude of soil N correct. In these models, there is N that is in slow cycling SOM pools that plants can't access. It would be more accurate to state that large scale ESM's don't include a mechanism whereby plants can access N bound in SOM as well as inorganic N.

Response: We have reformulated this sentence to reflect your insights.

Specifically line 37: "In contrast, N organically bound in soil organic matter (N-SOM), by far the largest ecosystem pool of soil N²⁰, is generally considered inaccessible to plants and is very rarely modelled to directly stimulate plant growth in ESM²¹"

Lines 89-90 and the final line of the manuscript: The idea that plants short circuit the N cycle and access organic nitrogen forms is not cryptic or relatively unrecognized. This is at least a twenty-year old idea and may even be older (see Schimel and Bennett 2004 Ecology and references there-in). It might be better to frame this as ecosystem scale evidence or something else along those lines.

Response: Thank you for pointing out the historical basis of this N cycling pathway. We have now provided additional context for these statements to address your concern and included this citation.

Specifically on line 40: "However, there is renewed interest in the possibility that acquisition of N-SOM may allow certain plants to "short-circuit" limiting supply rates of inorganic N^{22,23}"

Line 101-102: This language makes it read as if this present study is an explicit test. It is compelling evidence but it is not an explicit test of this idea. See large comment above.

Response: Here and elsewhere, we have rewritten such statements to reflect the inherently correlative nature of our study.

Line 231: This paper shows a CO₂ effect on tree rings:
Mathias, Justin M., and Richard B. Thomas. "Disentangling the effects of acidic air pollution, atmospheric CO₂, and climate change on recent growth of red spruce trees in the Central Appalachian Mountains." *Global change biology* 24, no. 9 (2018): 3938-3953.

Response: Thank you for this citation recommend. We have now included it in the manuscript, and have more thoroughly discussed previous literature on this subject. See

Line 68 in the present draft.

Line 245: I understand the need to highlight the importance of the findings here but suggesting that growth responses to elevated CO₂ for ECM trees are likely modest based on 54 chronologies is a bit of an overreach.

Response: We have rewritten this statement to note that our findings could be extrapolated to suggest that growth responses to eCO₂ may be modest and diverse for ECM associated trees.

Line 193: “Because the positive growth response to iCO₂ reported here is relativized and primarily occurred for the individuals with the smallest initial BAI, our study suggests that positive biomass responses to iCO₂ may be modest, and if they can be extrapolated, suggest overestimates for ECM tree response to iCO₂ at global scales^{28,32}”

Reviewer #3

I had been able to look at your submission "Ectomycorrhizal access to organic nitrogen mediates tree CO₂ fertilization response", which I consider a very thorough and comprehensive study. While I am not an expert in the field, I can say that the tree ring-related aspect of the study is sound and well done.

Response: Thank you for your assessment and review of the dendrochronological portions of our work

Reviewer #4

I found the manuscript by Pellitier et al. confusing, overly complicated in conceptual framework, and poor in methodological treatment. Even though they use a Bayesian approach, tree ring studies are not suitable to robustly detect changes driven by CO₂ because it is almost impossible to isolate the CO₂ signal from the effects driven by hundreds of other environmental factors and their interactions.

Response: We have rewritten large portions of the introduction, results, and discussion to improve the presentation and clarity of our findings.

- The discussion is poor. If I understood correctly, the main conclusion is that ECM plants use organic N when inorganic N is low. This seems pretty obvious. The other conclusion is that ECM plants do not increase their growth in response to CO₂ rising when inorganic N is high. This second conclusion contradicts probably the most well settled conclusion from hundreds of CO₂ experiments concluding that the CO₂ fertilization effect is high when N (inorganic N) is high (e.g. Reich et al. 2006, Norby et al. 2010, Norby & Zak 2011, Reich & Hobbie 2013). This seems to be the most important conclusion and yet it is not discussed by the authors. As a consequence, the only potentially surprising conclusion is simply ignored. This left me with the

impression that the conclusions are not justified due to biases in the method (see below), so the authors try to hide it.

Response: Thank you for encouraging us to improve the presentation and clarify of the discussion section. You will find an expanded discussion of our results in context with those you have specified above. First we would like to clarify that our analysis used “relativized measures of tree growth” allowing comparisons across the soil gradient; we used GNE (growth nitrogen efficiency) in our analysis to assess the relationship between increasing CO₂ and tree growth per unit of available inorganic N.

The four studies you mention artificially added inorganic N as mineral fertilizers. As a result, these studies are not directly comparable to our work. Foremost, field-based experiments that manipulate both eCO₂ and soil N often add ecologically unrealistic levels of fertilizer N, as in Reich et al. 2006. Although this is useful to experimentally understand how plants respond to limiting resources, it becomes tenuous when trying to place these results into an ecologically realistic context. Secondly, the addition of inorganic N to soils is known to strongly impact the composition and function of ectomycorrhizal communities (Lilleskov et al., 2002 *New Phytologist*; Lilleskov et al. 2002 *Ecology*), and generally reduces plant reliance on ECM symbionts for N uptake. Moreover, the addition of inorganic N is thought to reduce the decay capacity of ECM fungi (Lilleskov et al. 2002 *New Phytologist*; Bödeker et al., 2014 *New Phytologist*). As a result, the addition of inorganic N in these experiments is highly likely to have reduced ECM colonization, favored direct plant N uptake of the artificially added inorganic N and disfavored the N uptake pathway studied here: N-SOM. Our study allows initial investigations of ectomycorrhizal communities driving plant response to iCO₂ under a natural regime of soil N availability. We have discussed this limitation of soil N fertilization experiments on line 220.

We have explicitly discussed this “surprising conclusion” head on, and have distinguished how these experimental manipulations vary from the present study. We make the point that these previous studies actively preclude studying the proposed mechanism supported in our study (Line 203-222). We also note that the studies you mention, led by Peter Reich are conducted using pots in which grasses or forbs were studied. These arbuscular mycorrhizal associated plants are hypothesized to respond quite distinctly from ECM trees (Terrer et al. 2016; *Science*).

Terrer, C., Vicca, S., Hungate, B.A., Phillips, R.P. and Prentice, I.C., 2016. Mycorrhizal association as a primary control of the CO₂ fertilization effect. *Science*, 353(6294), pp.72-74.

Lilleskov, E.A., Fahey, T.J., Horton, T.R. and Lovett, G.M., 2002. Belowground ectomycorrhizal fungal community change over a nitrogen deposition gradient in Alaska. *Ecology*, 83(1), pp.104-115.

Lilleskov, E.A., Hobbie, E.A. and Fahey, T.J., 2002. Ectomycorrhizal fungal taxa differing in response to nitrogen deposition also differ in pure culture organic nitrogen use and natural abundance of nitrogen isotopes. *New Phytologist*, 154(1), pp.219-231.

Bödeker, I.T., Clemmensen, K.E., de Boer, W., Martin, F., Olson, Å. and Lindahl, B.D., 2014. Ectomycorrhizal *C. ortinarius* species participate in enzymatic oxidation of humus in northern forest ecosystems. *New Phytologist*, 203(1), pp.245-256.

-Indeed, a number of studies have found no evidence in tree-ring widths for any positive effect of CO₂ on growth. However, the methods used to test for long-term trends in tree-ring records are flawed and “may introduce considerable biases in quantifications of forest responses to environmental change” (Nehrbass-Ahles et al. 2014 GCB). See also Brienen et al. 2012. This is vociferously acknowledged by the tree-ring community, and they recognize more work is needed to make tree-ring data useful as a method to study trends in plant growth with environmental drivers while accounting for confounding factors (e.g. tree age, size). As a consequence of this bias, findings of no growth stimulation with rising CO₂ (e.g. van der Sleen et al. 2014) are common, but inconsistent with all other evidence (Walker et al. 2020).

Response: We do not discount that tree-ring studies must be carefully analyzed, and like many techniques, have inherent biases. See line 68 and 262 in the revised manuscript. However, tree-ring studies are widely acknowledged as representing a robust approach to empirically estimate the extent of the CO₂ fertilization response (see Gedalof and Berg 2010; Global Biogeochemical Cycles). In fact, Walker et al. 2021, has reviewed numbers of existing studies that employ dendrochronological approaches to study the stimulatory effects of CO₂ on tree growth. These dendrochronological techniques continue to represent an important component of the assessment of forest response to iCO₂ (see “high priority recommendations” as advocated in Walker et al. 2021). Walker et al., also highlight several papers from both temperate and boreal forest ecosystems that exhibit a positive growth response to elevated and increasing CO₂. Moreover, a widely cited review (Gedalof and Berg 2010), notes that approximately 20% of studied research sites “exhibit increasing trends in growth that cannot be attributed to climatic causes, nitrogen deposition, elevation, or latitude, which we attribute to a direct CO₂ fertilization effect”.

Directly quoting Walker et al. 2021: “Tree-ring analysis at CO₂ springs in Italy (two sites) suggests that eCO₂ increased *Quercus ilex* tree-ring width (a proxy for wood BP) initially ($\beta_{app} = 0.49-0.81$), and the increase diminished as trees aged (Hättenschwiler *et al.*, 1997). Basal-area increment (BAI) analysis showed the eCO₂ response stabilized at around 10 yr ($\beta_{app} = 0.27$) (Norby *et al.*, 1999). A large number of tree-ring studies have found little evidence for increases in wood BP. No detectable trends in BAI were found across tropical forests (3 sites, 12 species) (van der Sleen *et al.*, 2015), and both increasing and decreasing trends were found across North American boreal forests (598 sites, 19 species) (Girardin *et al.*, 2016). Syntheses across biomes found no significant increase in tree-ring width since 1950 ($\beta_{app} = 0.23 \pm 0.8$; 40 sites) (Peñuelas *et al.*, 2011) and variable responses of BAI ($\beta_{app} = 0.45 \pm 0.7$; 37 sites, 22 species) (Silva & Anand, 2013). Conversely, *Pinus* and *Quercus* tree rings from Missouri showed a positive response to iCO₂ that diminished with tree age ($\beta_{app} = 3.3$, at age 1 yr; $\beta_{app} = 1.1$, at age 50 yr) (Voelker *et al.*, 2006)”

The study mentioned (van der Sleen et al. 2015), is drawn from a tropical region. The majority of the trees studied in van der Sleen et al. (2015), associate with arbuscular

mycorrhizal fungi, which results in the tree being entirely dependent upon inorganic N for their growth. Therefore, incidentally, these findings may actually support available hypotheses regarding the CO₂ response of trees (ergo Terrer et al., 2016; Terrer et al. 2018). The dominance of AM trees in the tropics is increasingly quantified (Steidinger et al. 2019: Nature).

We are aware of limitations of employing tree-ring data, and have carefully accounted for these biases in the data and the variables included in the model. A different specialist reviewer found that the “tree ring-related aspect of the study is sound and well done”. We have explicitly mentioned the challenge of using dendrochronological approaches to infer the eCO₂ effect and have tempered our language in the introduction section (see below comments).

Gedalof, Z.E. and Berg, A.A., 2010. Tree ring evidence for limited direct CO₂ fertilization of forests over the 20th century. *Global Biogeochemical Cycles*, 24(3).

Hättenschwiler, S., Miglietta, F., Raschi, A. and Körner, C., 1997. Thirty years of in situ tree growth under elevated CO₂: a model for future forest responses?. *Global Change Biology*, 3(5), pp.463-471.

Norby, R.J., DeLucia, E.H., Gielen, B., Calfapietra, C., Giardina, C.P., King, J.S., Ledford, J., McCarthy, H.R., Moore, D.J., Ceulemans, R. and De Angelis, P., 2005. Forest response to elevated CO₂ is conserved across a broad range of productivity. *Proceedings of the National Academy of Sciences*, 102(50), pp.18052-18056.

Van Der Slepen, P., Groenendijk, P., Vlam, M., Anten, N.P., Boom, A., Bongers, F., Pons, T.L., Terburg, G. and Zuidema, P.A., 2015. No growth stimulation of tropical trees by 150 years of CO₂ fertilization but water-use efficiency increased. *Nature geoscience*, 8(1), pp.24-28.

Peñuelas, J., Canadell, J.G. and Ogaya, R., 2011. Increased water use efficiency during the 20th century did not translate into enhanced tree growth. *Global Ecology and Biogeography*, 20(4), pp.597-608.

Silva, L.C. and Anand, M., 2013. Probing for the influence of atmospheric CO₂ and climate change on forest ecosystems across biomes. *Global Ecology and Biogeography*, 22(1), pp.83-92.

Voelker, S.L., Muzika, R.M., Guyette, R.P. and Stambaugh, M.C., 2006. Historical CO₂ growth enhancement declines with age in *Quercus* and *Pinus*. *Ecological Monographs*, 76(4), pp.549- 564.

Terrer, C., Vicca, S., Hungate, B.A., Phillips, R.P. and Prentice, I.C., 2016. Mycorrhizal association as a primary control of the CO₂ fertilization effect. *Science*, 353(6294), pp.72-74.

Terrer, C., Vicca, S., Stocker, B.D., Hungate, B.A., Phillips, R.P., Reich, P.B., Finzi, A.C. and Prentice, I.C., 2018. Ecosystem responses to elevated CO₂ governed by plant-soil interactions and the cost of nitrogen acquisition. *New Phytologist*, 217(2), pp.507-522.

I believe these issues should not prevent tree-ring analyses from being published. The impact of such analyses for CO₂ effects is however doubtful, unless these biases are acknowledged and dealt with in an innovative way. To a bare minimum, these issues should be acknowledged in the intro, and the language should be humbler given the limitations. I don't support the bold language used in the intro, equating the validity of CO₂ experiments and tree-ring analyses in quantifying CO₂ effects.

Response: We have explicitly mentioned the challenge of using dendrochronological approaches to infer the eCO₂ effect and have tempered our language in the introduction section. For example, we have made this statement clear in line 68, and cited (Nehrbass-Ahles et al. 2014 GCB): “Although not without methodological limitations⁴¹, dendrochronological studies can be used to study historical responses to iCO₂^{6,14-16,42}”.

Moreover, we have again discussed such limitations in line 262 in the discussion section: “Finally, we acknowledge that there are considerable biases that affect the interpretation of dendrochronological data; importantly, our relativized analyses and use of a standardized host at small geographic scales reduces some of these limitations⁴¹.”

- Their hypothesis doesn't make any sense to me: “we reason that if N-SOM is critical to a positive eCO₂ fertilization effect, then trees associating with ECM taxa with greater decay capacity, i.e. occurring in N poor soils, will exhibit the largest relative fertilization response to eCO₂ (Fig. 1). In contrast, we predict that trees that primarily obtain inorganic N, i.e. occurring in inorganic N rich soils, will exhibit a reduced growth response to eCO₂ (Fig. 1)”. To my understanding, N limits the capacity of plants to convert eCO₂ to additional growth. It doesn't matter if the N comes from inorganic or organic sources. In any case, using organic sources would decrease growth, not increase. The reason is that, as the authors describe in the intro, plant organic N uptake requires a carbon investment in ECM fungal mining of N, which reduces the amount of carbohydrates that can be allocated to plant growth. I am puzzled with this hypothesis, which is the central argument of the paper. You can look at it from a different perspective: why would the CO₂ effect be larger under low than high N availability? It doesn't make any sense.

Response: Thank you for encouraging us to improve the clarity of our reasoning and hypotheses. Terrer et al., 2016 Science, were among the first to posit that the capacity of ECM fungi to obtain N-SOM could result in the capacity of their host trees to respond positively to increasing CO₂. This is because natural supply rates of inorganic N (in the *absence* of artificial inorganic N addition), are insufficient to engender a positive plant growth response to CO₂. These results may seem paradoxical and apparently conflict with previous literature at first glance; however, they may be parsimonious. Trees growing under natural soil conditions with intermediate or high natural availabilities of inorganic N may not associate with ECM communities that can obtain N-SOM. Artificial addition of inorganic N, however, would likely uniformly increase their growth response to eCO₂, irrespective of initial soil N conditions. Because our analyses are standardized to unit of inorganic N availability as noted in line 64, 193 and elsewhere, the capacity of certain trees to obtain N-SOM via ECM symbionts, may serve to ‘enhance’ overall plant N uptake. In fact, greater progressive allocation of C to specialized ECM fungi capable of obtaining N-

SOM, (as predicted by Terrer et al. 2018: *New Phytologist*), is likely to increase overall plant N uptake. However, this effect would only occur for a subset of the trees studied here (i.e. occurring in low inorganic N soils). We have directly mentioned this overall logic in the discussion section, Lines 203-222.

Terrer, C., Vicca, S., Stocker, B.D., Hungate, B.A., Phillips, R.P., Reich, P.B., Finzi, A.C. and Prentice, I.C., 2018. Ecosystem responses to elevated CO₂ governed by plant–soil interactions and the cost of nitrogen acquisition. *New Phytologist*, 217(2), pp.507-522.

Terrer, C., Vicca, S., Hungate, B.A., Phillips, R.P. and Prentice, I.C., 2016. Mycorrhizal association as a primary control of the CO₂ fertilization effect. *Science*, 353(6294), pp.72-74.

- Fig 1: similar to my comment right above, I don't understand the reasoning in Fig. 1. The figure is extremely confusing. The tree with the widest CO₂ arrow (representing larger CO₂ fertilization effect) is represented by the smallest tree size. Shouldn't be the opposite? Also, the CO₂ arrows give the impression they refer to CO₂ concentration

Response: We have made the tree 'sizes' identical across the depicted gradient. We also want to reiterate the point made in the text that even if trees growing under low inorganic N are the ones benefiting most from iCO₂, these responses are relativized. Such individuals still have low absolute growth rates because overall their overall N budget (organic and inorganic) is potentially smaller than those of trees growing at high inorganic N (even if these only have inorganic N available).

See Line 193 for our discussion in the main text: “Because the positive growth response to iCO₂ reported here is relativized and primarily occurred for the individuals with the smallest initial BAI, our study suggests that positive biomass responses to iCO₂ may be modest, and if they can be extrapolated, suggest overestimates for ECM tree response to iCO₂ at global scales^{28,32}”.

- Intro: the statement linking the (lack of) response in mature eCO₂ experiments with results from tree-ring analyses is misleading; this statement suggests there is no such thing as a CO₂ fertilization effect, as corroborated by both mature experiments and tree-ring analysis. This is incorrect. The vast majority of sources of evidence suggest that CO₂ increases carbon uptake. Tree-ring studies are the exception (e.g. Walker et al. 2020 *New Phyt*). It is true that the few CO₂ experiments with mature plants show no growth stimulation. However, there are just a handful or less of examples, and all of them occur in extreme nitrogen (Sigurdson et al.) or phosphorus (Ellsworth et al, Jiang et al) limitations and results from these experiments cannot be extrapolated to other ecosystems (comment by Luo & Niu 2020).

Response: We have clarified these statements in the Introduction, and included the recent article (Jiang et al. 2020 *Science*), to illustrate the relatively minor stimulatory effects of CO₂ in mature ecosystems. Walker et al., 2021 note that the majority of studies are biased because they occur in young ecosystems, accordingly the present study occurring in a late-successional ecosystem provides unique insights. Because we are primarily discussing

mature ecosystems, and thus, we agree with your statement that “the few CO₂ experiments with mature plants show no growth stimulation”. You will find that we have drawn explicit attention to these contrasting results from young and mature forest ecosystems. In particular, please see Lines 32 and 196.

Line 32: “These modest growth responses often stand in contrast to those derived from early successional ecosystems^{10,12,17}, suggesting that the stimulatory effects of CO₂ may be transient.

Line 196: “ In fact our findings ranging from a modest to neutral growth response in aggregate, are in line with available evidence for eCO₂ experiments conducted in mature ecosystems⁷, further highlighting the importance of nutrient limitation and that studies derived from aggrading or early successional ecosystems may be poor predictors of forest C sequestration under future CO₂ regimes¹².

Jiang, M., Medlyn, B.E., Drake, J.E., Duursma, R.A., Anderson, I.C., Barton, C.V., Boer, M.M., Carrillo, Y., Castañeda-Gómez, L., Collins, L. and Crous, K.Y., 2020. The fate of carbon in a mature forest under carbon dioxide enrichment. *Nature*, 580(7802), pp.227-231.

The entire framework is too complicated, and the logic is hard to follow. For example, it took me a while to understand why the authors expect a relationship between BAI and N mineralization, which is the main foundation of their conclusions. Even after reading the paper, I am not too sure I follow the logic. I must say I have participated in over a dozen papers on the N cycle, and yet I struggled to follow. I doubt the topic, structure, methods and writing of the paper are suitable for a wide audience, as someone would expect in Nature Communications.

Response: We recognize that this is a complex system and we are not carrying out a controlled experiment, but instead we are analyzing natural data and making inferences about several interconnected processes. Figure 1 is our attempt to bring this all together. We have edited the introduction and results section to more clearly illustrate the rationale for our logic, as well as linkages between our statistical analyses and the key results and figures. This is the first time these processes have been linked together in a system, and we hope this work will spark similar types of integrated research in the future.

- L74 “models” x2

Response: Corrected

I don't understand why GPP is mentioned in the abstract and is the lead in the introduction. The authors focus on stem growth, not GPP. NPP would be closer, but they just measure BAI which is far from NPP.

Response: We have deleted any mention of GPP in the abstract. GPP is mentioned in the introduction because it represents the primary focus of many atmospheric monitoring and

modeling activities. We include it because it provides important context for the broader body of literature.

L75 there is no consensus about the statement “[...] ca. 2010”. That is just a result from a particular study and not a generalization. Stimulation of GPP by eCO₂ would continue as long as CO₂ continues increasing or the saturation point of photosynthesis is reached. Both points are uncertain, and the effect doesn’t stop in a particular year. Please remove because it gives a wrong impression.

Response: We have removed this statement.

Reviewer comments, second round -

Reviewer #2 (Remarks to the Author):

The authors did a good job at responding to my comments as well as the other reviewer comments. The revisions they have made have strengthened the manuscript.

Reviewer #4 (Remarks to the Author):

I had a quick read to the new manuscript, and I still found it much more difficult to follow than it should. Unfortunately, I still don't follow the logic, and the conceptual framework is not clear. Even the abstract is hard to decipher and extract anything useful. So far, I don't see the value of this manuscript to the field. I work actively in this field, and I can't picture myself citing this manuscript because I still don't know what's the research gap this manuscript is trying to fill. In fact, the authors cite one paper I have participated, but for the wrong reasons. I still don't see the fit for this journal, and I stick to my original review because I don't see much improvement in terms of value. Unfortunately, I don't have the time to improve the manuscript, so I will leave to the editor and reviewers for a decision.

Reviewer #5 (Remarks to the Author):

The authors studied how a tree species responds to increasing ambient CO₂ along a nitrogen mineralization gradient, by estimating yearly tree growth and characterizing root fungal communities. Their main conclusion is that tree growth response along the gradient depends on their association to different ectomycorrhizal fungi. The manuscript is well written and easy to follow conceptually, and represents a good contribution to subject matter, which is of timely importance. The authors also appear to have put considerable effort into restructuring their paper and making the necessary changes requested by the other reviewers, or provided in detail a suitable rebuttal. The methods used seem appropriate, although I cannot comment on the dendrochronology and bayesian inference. I have some comments and suggestions regarding the analysis and interpretation that may help to further improve the paper.

First it would be good if the authors could discuss the potential role of other microbial groups other than ECM fungi. Even within ECM fungi, the authors only focused on two genera, and I could not find a presentation of the whole ECM fungal community or exploration of the other ECM taxa beyond Cortinarius and Russula.

I think the authors cannot exclude the possibility that the observed association could be because of niche preferences of ECM fungi to different N availability across the gradient. It would be good if the authors could discuss cause-and-effect relationships and whether there could be some confounding variables such as niche preferences of ECM fungi along the gradient.

Since the authors studied only one species, can they discuss how their data can be generalized to other plant families or functional groups? Are there not some growth traits in Quercus that may make it distinct from other species?

The authors need to better describe how they dealt with spatial autocorrelation in their data.

It was also unclear how the authors accounted for library size differences across their metagenomics samples.

Specific points

L55- L57 How about the priority effect across your study sites? Isn't it possible that some non-preferential ECM fungi establish themselves early across the gradient, which may make it difficult for the trees to associate with the preferential ones? This also makes me wonder about the cause-and-effect relationship between ECM fungi and mineralization rates. Would not it be possible that

the observed differences between the composition ECM fungi at least to some extent relate to their niche differentiation across the gradient rather than their functions?

L59- L60 The authors could emphasize that not only host but also environmental conditions and interactions between fungi could determine which partners associate with the host.

L77 On what basis your study sites were selected across the gradients?

L96 & L348 Does this analysis account for spatial autocorrelation among their sites? The clustered distribution of some of the sites (Fig. S1) suggests that the authors need to account for this.

Please clarify.

L111 & Fig. S3. Why not showing a ordination plot showing how the composition of ECM fungal taxa or morphotypes change across the gradient?

L117 What proportion of ECM roots and sequences belonged to these two groups (Cortinarius and Russula)? Why focusing only on these two groups? As a reader I would like to see some hints to this in the introduction.

L141- L144 Was this compositional shift limited to ECM decay genes only? Could the authors for example examine other gene groups, for example those related to N mineralization or N cycle? Fig. 3A,B. Assuming that these two groups are the dominant groups of ECM fungi in your dataset, how the authors can exclude the possibility that it is not only Cortinarius that is changing across the gradient and that would artificially affect the relative abundance of Russula (or vice versa)? I have a similar concern regarding Fig. 3C,D.

L212-L216 I would not necessary agree that the activity of of free living saprotrophs is independent of ECM fungi, I think the authors should consider ideas around the potential competition between ECM and saprotrophs for organic N, i.e. the Gadgil effect, and how this may relate to tree N acquisition.

Fig.3E & L136 I did not find F and P values for the summed abundance of CAZy.

Fig. 3. E-F The distribution of counts overlaps between below and above the change point for all genes, and the differences seem to be marginally significant according to Table S2. Why only showing these genes, when there are some GH genes showing greater differences (based on Table S2)? Furthermore, did the authors normalize the data based on the library size per sample?

Fig. S2a, Could data below and above the change point be indicated?

Fig. S6 The pattern does not seem particularly strong.

Fig.S8B The composition changes but it is unclear to me whether the potential for decomposition changes across the gradient.

Table S3. How did the authors include geographic distance in their models? I could not find this information in the methods.

Minor points

L1 I am not sure opening with plant-microbe interactions represents this paper and its content, perhaps plant-mycorrhizal interactions is more accurate as you only look at this particular group of "microbes".

L6 ectomycorrhizal >> ectomycorrhizal (ECM)

L48 Perhaps the authors could emphasise that this is relevant mainly to temperate forests.

L58 Change to ECM fungal taxa.

L76 Might be useful to specify the range for the gradient here?

L83 Could the authors briefly describe what the method does and why this method here?

L236-L239 What do the authors mean by isotopically depleted or enriched foliage? Could they please be more specific i.e. ^{15}N enriched or depleted foliage, or foliage depleted or enriched in isotopically heavy N. Also I think a brief mention of the potential mechanism that supports that the ^{15}N status of the foliage relates to a transition from organic to inorganic N assimilation is warranted here.

Reviewer #5 (Remarks to the Author):

The authors studied how a tree species responds to increasing ambient CO₂ along a nitrogen mineralization gradient, by estimating yearly tree growth and characterizing root fungal communities. Their main conclusion is that tree growth response along the gradient depends on their association to different ectomycorrhizal fungi. The manuscript is well written and easy to follow conceptually, and represents a good contribution to subject matter, which is of timely importance. The authors also appear to have put considerable effort into restructuring their paper and making the necessary changes requested by the other reviewers, or provided in detail a suitable rebuttal. The methods used seem appropriate, although I cannot comment on the dendrochronology and bayesian inference. I have some comments and suggestions regarding the analysis and interpretation that may help to further improve the paper.

First it would be good if the authors could discuss the potential role of other microbial groups other than ECM fungi. Even within ECM fungi, the authors only focused on two genera, and I could not find a presentation of the whole ECM fungal community or exploration of the other ECM taxa beyond *Cortinarius* and *Russula*.

Response: Based on your recommendations we have discussed the possibility for distinct saprotrophic communities. Within the ECM guild, we present the full spectrum of ECM communities in Figure S4. In our revisions we have now discussed in brief the other ECM members that were present across the gradient such as *Hebeloma* in the Results section, and directed readers to the manuscript Pellitier et al., 2021 Ecosystems.

I think the authors cannot exclude the possibility that the observed association could be because of niche preferences of ECM fungi to different N availability across the gradient. It would be good if the authors could discuss cause-and-effect relationships and whether there could be some confounding variables such as niche preferences of ECM fungi along the gradient.

Response: We are happy to provide a deeper discussion of plausible community assembly processes that give rise to the observed patterns in ECM communities studied here. To do so, we have added the following text in the revised Discussion section (Line 198): “If the possession of such traits is associated with enhanced photosynthate allocation and persistence on host root-tips, these traits may be especially favored under conditions of low inorganic N availability³⁸

Since the authors studied only one species, can they discuss how their data can be generalized to other plant families or functional groups? Are there not some growth traits in *Quercus* that may make it distinct from other species?

Response: *Quercus rubra* is a dominant ectomycorrhizal host tree throughout eastern North America, and therefore our study has wide applicability to the surrounding region. Moreover, *Quercus* is distributed across several continents, including in some portions of the tropics. Most ectomycorrhiza are host generalists, and few ECM taxa show high specificity with *Quercus*. In this sense, while we

cannot guarantee that our results would be identical if a different host was used, we do not expect the identity of the host to have an outside influence on the patterns observed here. We are unaware of specific life-history differences between *Quercus* or a *Pinaceae* that would alter our findings.

To address this comment we have mentioned in the Discussion section Line 248: “Further studies on a wider range of ECM and AM communities and host trees, distributed across forest biomes, particularly the tropics, are needed to determine the generality of our findings”

The authors need to better describe how they dealt with spatial autocorrelation in their data.

Response: Lines 93, and 159 of the Results section describe in brief that spatially explicit random effects were incorporated into the Bayesian models incorporating dendrochronological measurements. Model terms and overall approach are described in full in the *Methods* section and *Supplementary Methods*. Finally a GDM used to assess shifts in the compositional abundance of ECM decay genes ECM uses a spatially explicit autocorrelative structure which is described in the *Methods*.

It was also unclear how the authors accounted for library size differences across their metagenomics samples.

Response: This information has been more fully described in the revisions. Foremost, differences in the abundance of quality filtered reads did not significantly vary across samples (Figure S12). Nevertheless, we explicitly accounted for library size differences and the compositional nature of the datasets. Specifically, “to account for the compositional nature of the metagenomic data, we calculated the logarithm of the number of sequences mapped to a given decay gene family divided by the geometric mean number of orthologous near single-copy features present in the sample (single-copy genes). Note that this is identical to an additive log-ratio transformation (Quinn *et al.*, 2018). This reveals how the relative abundance of decay genes present in a community behave relative to the estimate of the number of single-copy genes (genomes) present in each sample”.

Quinn, T.P., Erb, I., Richardson, M.F. and Crowley, T.M., 2018. Understanding sequencing data as compositions: an outlook and review. *Bioinformatics*, 34(16), pp.2870-2878.

Specific points

L55- L57 How about the priority effect across your study sites? Isn't it possible that some non-preferential ECM fungi establish themselves early across the gradient, which may make it difficult for the trees to associate with the preferential ones? This also makes me wonder about the cause-and-effect relationship between ECM fungi and mineralization rates. Would not it be possible that the observed differences between the composition ECM fungi at least to some extent relate to their niche differentiation across the gradient rather than their functions?

Response: Assembly processes that structure ectomycorrhizal communities remain a key area of study (Bogar and Peay 2017). Priority effects have been demonstrated for ECM fungi and we do not discount their potential to impact the patterns documented here. A key observation of studies investigating priority effects in open-communities is that the strength of priority attenuates through time (i.e Fukami 2015. Annual Rev. In the study system described here, plant and fungal communities have assembled following the last glacial maxima ~ 10,000 years ago. Although the forest ecosystems studied here were clearcut in the early 1900's, clear-cutting was conducted in winter, and root-systems were therefore not disturbed during this process. Many resprouts of *Quercus rubra* are observed in the region. Finally, we agree that niche differences among ECM taxa are also likely to be impacted by functional differentiation and trade-offs in nutrient acquisition or other functional axes.

In an effort to address your comment, we have rewritten these sentences to accommodate a broader set of assembly processes, and deemphasized the statement that 'trees preferentially partner with ECM that maximize N return on C expenditure'. Moreover, we have rewritten Line 55, as follows: "Biological market perspectives emphasizing the metabolic cost of fungal resource capture³³ suggest that trees may associate with ECM mutualists that maximize N acquisition and minimize plant carbon (C) expenditure (N return on C investment)". We also acknowledge 'niche based processes on line 194 in the Discussion.

Bogar, L.M. and Peay, K.G., 2017. Processes maintaining the coexistence of ectomycorrhizal fungi at a fine spatial scale. In *Biogeography of mycorrhizal symbiosis* (pp. 79-105). Springer, Cham.

L59- L60 The authors could emphasize that not only host but also environmental conditions and interactions between fungi could determine which partners associate with the host.

Response: We have address this comment by noting in the Introduction that a broad array of assembly processes impact ECM communities, citing Bogar and Peay (2017; see above). Moreover, based on your comment, we have rewritten certain elements of the Discussion to address this point (i.e. Line 196-202).

L77 On what basis your study sites were selected across the gradients?

Response: Sites were randomly selected from a total of 58 sites that were subdivided into ' three ecosystem types' in order to span semi-continuously the entirety of the soil mineralization gradient. The total 58 sites were originally compiled from previous ecosystem classification efforts conducted in the region (Barnes et al., 1982) and employed by the senior author (Zak et al., 1990).

Barnes, B.V., Pregitzer, K.S., Spies, T.A. and Spooner, V.H., 1982. Ecological forest site classification. *Journal of Forestry*, 80(8), pp.493-498.

Zak, D.R. and Pregitzer, K.S., 1990. Spatial and temporal variability of nitrogen cycling in northern Lower Michigan. *Forest Science*, 36(2), pp.367-380.

L96 & L348 Does this analysis account for spatial autocorrelation among their sites? The clustered distribution of some of the sites (Fig. S1) suggests that the authors need to account for this. Please clarify.

Response: Yes, spatial autocorrelation is accounted for in this and additional analyses. We have specifically discussed how spatial autocorrelation was accounted for in the above response.

L111 & Fig. S3. Why not showing a ordination plot showing how the composition of ECM fungal taxa or morphotypes change across the gradient?

Response: We have now added an ordination as you describe in Figure S4.

L117 What proportion of ECM roots and sequences belonged to these two groups (*Cortinarius* and *Russula*)? Why focusing only on these two groups? As a reader I would like to see some hints to this in the introduction.

Response: The proportion of sequences ascribed to *Cortinarius* and *Russula* is presented in Figure 3A/3B. These two ECM genera are globally widespread and here comprise the majority of ECM sequences detected (sum of >55% of sequences across the gradient). In our revisions we briefly discussed the life history attributes of *Cortinarius* in the Introduction (Line 60). Moreover, in the Methods section, we have directed readers to supplementary Figure describing key taxa. In the Results section we have also described the presence of *Hebeloma* in low certain low inorganic N soils.

L141- L144 Was this compositional shift limited to ECM decay genes only? Could the authors for example examine other gene groups, for example those related to N mineralization or N cycle?

Response: Unfortunately we have not conducted analyses related to the N cycle or N mineralization. This is due to the fact that these processes (N mineralization in particular) are dependent upon the free-living saprotrophic bacterial and fungal communities. Because we exclusively sampled ECM root-tips (to obtain proper metagenomic measures of ECM function), we do not have metagenomic sequences that effectively capture saprotrophic guilds in the soil that would contribute to mineralization rates.

Fig. 3A,B. Assuming that these two groups are the dominant groups of ECM fungi in your dataset, how the authors can exclude the possibility that it is not only *Cortinarius* that is changing across the gradient and that would artificially affect the relative abundance of *Russula* (or vice versa)? I have a similar concern regarding Fig. 3C,D.

Response: Such concerns represent a key challenge inherent to compositional datasets that rely upon relative abundances of microbial communities or functions.

Our datasets are not different, and it is true that the shifts in the relative abundance presented in Figure 3A,B and 3C,D can be misleading if they are meant to infer shifts in absolute abundances. To address your comments, we have clarified throughout our manuscript that we are comparing relative abundances of ECM communities, and stated in the Discussion (Line 211-213): “Additionally, metagenomic estimates of absolute functional gene abundance may provide deeper insight into the N-cycling pathways and mechanisms proposed here”

L212-L216 I would not necessarily agree that the activity of free living saprotrophs is independent of ECM fungi, I think the authors should consider ideas around the potential competition between ECM and saprotrophs for organic N, i.e. the Gadgil effect, and how this may relate to tree N acquisition.

Response: We agree that there is potential interactions between ECM fungi and free-living saprotrophs. However, as we did not explicitly measure Gadgil effects or other interactions we can only speculate as to their relative importance here. To address your comment we have included a sentence describing the potential importance for Gadgil type interactions to modify ECM and saprotrophic community activity, along with relevant citations. See line 213: “Future studies studying co-occurring saprotrophic communities may reveal interesting inter-guild interactions that structure plant access to N and SOM dynamics⁶⁴”.

Fig.3E & L136 I did not find F and P values for the summed abundance of CAZy.

Response: These have now been presented in the main text.

Fig. 3. E-F The distribution of counts overlaps between below and above the change point for all genes, and the differences seem to be marginally significant according to Table S2. Why only showing these genes, when there are some GH genes showing greater differences (based on Table S2)? Furthermore, did the authors normalize the data based on the library size per sample?

Response: The presentation of these specific gene families was based on knowledge of their biological function and general extracellular role in ectomycorrhizal decay of SOM. For example AA9,10,11 encode lytic polysaccharide monooxygenases that have been previously implicated in ECM decay of SOM (Kohler et al., 2015). Moreover, AA3_1 encodes cellobiose dehydrogenases which catalyze the production of H₂O₂ necessary LPMO activity (Janusz et al., 2017).

In an effort to present a more balanced description of certain key gene families that may underlie the tree growth responses presented here, we have provided a deeper description of key ECM gene families that may be involved in decay of SOM in the results and discussion section. Moreover we have also discussed that certain CAZy gene families, despite exhibiting significant differences above and below the change-point may encode alternative intracellular functions that may not be definitively related to SOM decay. Lines 203-205

Regarding library size normalization, we have discussed this concern in full in an above comment.

Kohler, A., Kuo, A., Nagy, L.G., Morin, E., Barry, K.W., Buscot, F., Canbäck, B., Choi, C., Cichocki, N., Clum, A. and Colpaert, J., 2015. Convergent losses of decay mechanisms and rapid turnover of symbiosis genes in mycorrhizal mutualists. *Nature genetics*, 47(4), pp.410-415.

Grzegorz Janusz, Anna Pawlik, Justyna Sulej, Urszula Świdorska-Burek, Anna Jarosz-Wilkolazka, Andrzej Paszczyński, Lignin degradation: microorganisms, enzymes involved, genomes analysis and evolution, *FEMS Microbiology Reviews*, Volume 41, Issue 6, November 2017, Pages 941–962, <https://doi.org/10.1093/femsre/fux049>

Fig. S2a, Could data below and above the change point be indicated?

Response: This graph depicts the relationship between predicted and observed BAI for model evaluation purposes only. The addition of points colored by change-point is unlikely to provide additional information as overall model-fit is very high and varies negligibly across the breadth of BAI size-classes.

Fig. S6 The pattern does not seem particularly strong.

Response: Based on the separation observed along PCA axis 1, this pattern is robust and confirmed using GDM. This ordination is solely intended as supporting information for visualization purposes.

Fig.S8B The composition changes but it is unclear to me whether the potential for decomposition changes across the gradient.

Response: This figure alone does not allow for the interpretation of greater decay potential below the change point. When considered in combination with Figure S8, and Figure 3 in the main-text, there is a greater relative abundance of key gene families implicated in the decay of SOM below the statistical change point.

Table S3. How did the authors include geographic distance in their models? I could not find this information in the methods.

Response: We direct the reviewer to Line 357 and 437 in the Methods. A fuller description of spatial sampling structure in GDM is described in full in Ferrier et al., 2007.

Ferrier, S., Manion, G., Elith, J. and Richardson, K., 2007. Using generalized dissimilarity modelling to analyse and predict patterns of beta diversity in regional biodiversity assessment. *Diversity and distributions*, 13(3), pp.252-264.

Minor points

L1 I am not sure opening with plant-microbe interactions represents this paper and its content, perhaps plant-mycorrhizal interactions is more accurate as you only look at this particular group of “microbes”.

Response: We do not disagree that mycorrhiza represents only a small fraction of microbes involved in soil N cycling. In our revisions we have rewritten this L1.

L6 ectomycorrhizal >> ectomycorrhizal (ECM)

Response: Corrected

L48 Perhaps the authors could emphasise that this is relevant mainly to temperate forests.

Response: We have chosen to not describe the relevancy to temperate forests, as the underlying mechanisms for ECM associated plant access to N-SOM are likely to be equivalent wherever such plants occur (i.e. boreal, temperate or tropical).

L58 Change to ECM fungal taxa.

Response: In an effort to maintain clarity, and as ECM are already defined as fungi above, we have not incorporated this revision.

L76 Might be useful to specify the range for the gradient here?

Response: We have described the range as ‘broad’ and refer readers to the Methods section for a detailed description.

L83 Could the authors briefly describe what the method does and why this method here?

Response: A fuller accounting of the model description and selection is described in Figure 1, as well as the *Methods* section.

L236-L239 What do the authors mean by isotopically depleted or enriched foliage? Could they please be more specific i.e. ^{15}N enriched or depleted foliage, or foliage depleted or enriched in isotopically heavy N. Also I think a brief mention of the potential mechanism that supports that the ^{15}N status of the foliage relates to a transition from organic to inorganic N assimilation is warranted here.

Response: We have rewritten this sentence as follows to address your comment: “Additional evidence for a transition from assimilation of N-SOM to predominately inorganic N comes from a previous study of these individuals trees, which found $\delta^{15}\text{N}$ depleted foliage under conditions of low inorganic N availability suggestive of enhanced organic N uptake, and relatively enriched foliage under high inorganic N availability:

Reviewer comments, third round -

Reviewer #5 (Remarks to the Author):

The authors have addressed the main concerns I raised in the previous report, and the manuscript has significantly improved.